# Estimation of gross land-use change and its uncertainty using a Bayesian data assimilation approach

*Peter Levy, Marcel van Oijen, Gwen Buys, and Sam Tomlinson*

*2018-02-02*

## Abstract

We present a method for estimating land-use change using a Bayesian data assimilation approach. The approach provides a general framework for combining multiple disparate data sources with a simple model. This allows us to constrain estimates of gross land-use change with reliable national-scale census data, whilst retaining the detailed information available from several other sources. Eight different data sources, with three different data structures, were combined in our posterior estimate of land-use and land-use change, and other data sources could easily be added in future. The tendency for observations to underestimate gross land-use change is accounted for by allowing for a skewed distribution in the likelihood function. The data structure produced has high temporal and spatial resolution, and is appropriate for dynamic process-based modelling. Uncertainty is propagated appropriately into the output, so we have a full posterior distribution of output and parameters. The data are available in the widely used netCDF file format from http://eidc.ceh.ac.uk/ (doi pending).

# Introduction

Human-induced land-use change has a substantial impact on biodiversity and both biogeo-chemical and hydrological cycles (Gitz and Ciais, 2003; Levy et al., 2004; Newbold et al., 2015; Piano et al., 2017; Post and Kwon, 2000). The importance of representing it in models of the climate, hydrology, and ecosystem processes is increasingly recognised (Martin et al., 2017; Prestele et al., 2017; Quesada et al., 2017). However, although changes in land use tend to occur incrementally over small areas, data on land-use change are typically limited in spatial and temporal resolution (Alexander et al., 2017). Furthermore, changes in land use may be rotational or involve transitions between multiple land-use classes over time, such that the gross area undergoing land-use change may be much larger than the net change in area (Fuchs et al., 2015; Tomlinson et al., 2017). From the point of view of modelling ecosystem processes, it is these fine-scale gross changes that we need to represent, because as model inputs, these may give very different simulated output, compared with simulations based on the net change at a coarse scale (Fuchs et al., 2015; Kato et al., 2013; Wilkenskjeld et al., 2014). For example, a reported net increase in forest area of 10 km$^2$ may actually result from afforestation of 50 km$^2$ and deforestation of 40 km$^2$. As input data to an ecosystem model, this might produce quite different results, compared to the parsimonious assumption (afforestation of 10 km$^2$ and no deforestation)(Krause et al., 2016; Levy and Milne, 2004). Over most of the globe, data on land-use change are typically limited in spatial and temporal resolution, and are typically represented by a time series of the area occupied by each land-use class (Rounsevell et al., 2006). Little information is available on the gross changes which bring about this time series (Prestele et al., 2017). The IPCC Good Practice Guidelines recommends the estimation of land-use change matrices for reporting GHG fluxes arising from land-use change (Penman et al., 2003). This provides explicit information on the areas which have changed from each land-use class to every other class. Whilst these matrices contain more information, they are only valid over the single time period for which they were

derived, being a two-dimensional summary. For modelling over longer time periods, these are not very useful in themselves. To properly represent the change in land use over time, we need a higher-dimensional data structure.

Land-use change is not easy to measure. A key problem is identifying change from repeated map or survey data, where the magnitude of the change signal is very small against the background noise of sampling and measurement error. Large censuses and careful survey techniques are required to distinguish true change from differences arising from measurement and sampling error (Fuller et al., 2003). A further problem is that information on land-use change at national scale typically comes from multiple disparate sources, which are often inconsistent with each other, using different land-use classifications and definitions (Phelps and Kaplan, 2017), arising from different thematic areas, and focus on different spatial and temporal domains, with different resolutions (Fisher et al., 2017). For example, land-use data in the UK are available from the agricultural census and surveys, the national forestry sector, the national mapping survey, as well as earth observation products such as Corine, MODIS and the CEH Land Cover Maps. However, no single data source provides a reliable estimate of land-use change with national coverage which extends suitably far back in time. A data assimilation approach is needed to make best use of the available data, so as to provide such a product. Existing methods ignore the large uncertainties which arise in estimating past land use change, and data assimilation approaches can explicitly address this issue.

In general terms, data assimilation is an approach for fusing observations with prior knowledge (e.g., mathematical representations of physical laws; model output) to obtain an estimate of the distribution of the true state of some phenomenon. It has become very commonly used in fields such as atmospheric and oceanographic modelling, and numerical weather prediction (e.g. Lunt et al., 2016). Various techniques are used, such as simulated annealing, ensemble Kalman filtering, and 4D variational assimilation. All of these can be seen as special cases within the Bayesian framework, where models, parameters and data are related in a formal

way via Bayes Theorem (Wikle and Berliner, 2007). There are some significant differences in applying data assimilation in our land-use context, compared with atmospheric modelling. Firstly, there is only a very simple model, compared with the complex physical models of the atmosphere or ocean. By contrast, the observational process by which the data are produced is extremely complex, compared with the simple observations of air or sea temperature or pressure. Also, we are predicting retrospectively (i.e. "hind-casting") over many years in the past, rather than "nudging" forecasts as new data becomes available.

Our aim here was to develop a generic Bayesian approach, using multiple sources of data, to make spatially- and temporally-explicit estimates of land-use change. In a case study, we apply the approach to Scotland over the period 1969-2015. As an example application, we use a simple model of carbon fluxes following land-use change to show how uncertainties surrounding land-use change can be propagated through to model output.

# Materials and methods

## Mathematical approach and notation

We represent land use $u$ as a number of discrete states from the set {forest, crop, grassland, roughgrazing, urba encoded as integers 1-6. At a single location (x,y), land use can change between these states over time, represented by the vector $\mathbf{U}_{xy}$. (We use a convention of representing matrices and arrays as uppercase bold (e.g. $\mathbf{U}$), and individual elements thereof as uppercase italic (e.g. $U_{xyt}$).) An example for $t = (1 \ldots 5)$ would be $\mathbf{U}_{xy} = (4, 3, 3, 2, 2)$, showing a change in land use from rough grazing (class 4) to grassland (class 3) for two years, then to cropland (class 2) for two years. Spatially, we represent land use on a grid, where each grid cell contains a vector of land use. Combining the spatial and temporal dimensions, we have the 3-D space-time array $\mathbf{U} = \{U_{xyt}\}$ (Figure 1). This is the basic data structure required by any model which models the effects of land use dynamically and spatially explicitly. Our aim is

to estimate the 3-D array **U** as accurately as possible by constraining with multiple data sources. (We note that for the purposes of non-spatial modelling, there is a lot of redundancy in this data structure, and the information in **U** can be condensed into the set of unique land-use vectors and their corresponding areas. We return to this point later.)

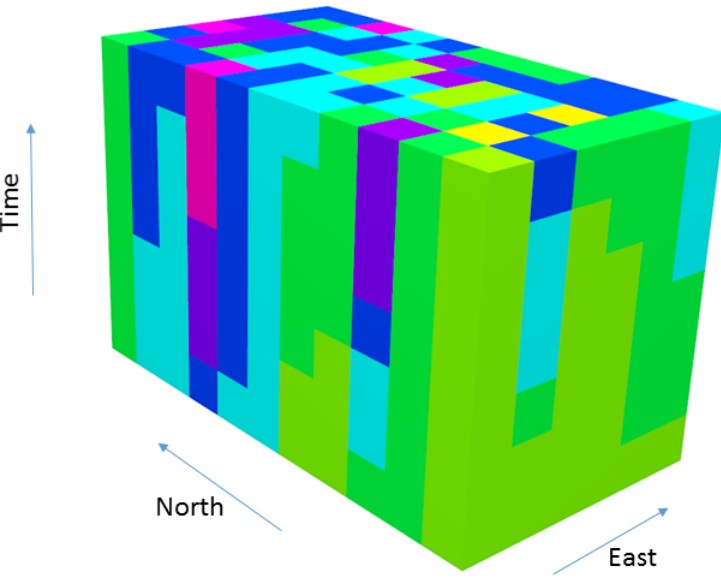

Figure 1: Graphical depiction of a hypothetical 3-D cuboid **U** representing land use in space and time dimensions. Different colours show different land uses.

We denote the area occupied by each land use $u$ at time $t$ as $A_{ut}$, obtained by counting the frequency of land uses in $\mathbf{U}_t$:

$$A_{ut} = \sum_{x=1}^{n_x} \sum_{y=1}^{n_y} [U_{xyt} = u] l^2 \tag{1}$$

where the square brackets are Iverson notation, evaluating to 1 where true and zero otherwise, and $l^2$ is the area of a single grid square. We denote the array of all these areas (for each

land-use class and time step) as $\mathbf{A} = \{A_{ut}\}$. By differencing, we obtain the areas of net land-use change:

$$\Delta A_{ut} = A_{ut} - A_{ut-1}. \tag{2}$$

At each time step, we have a square transition matrix

$$\mathbf{B} = \begin{bmatrix} 0 & \beta_{12} & \beta_{13} & \dots & \beta_{1n} \\ \beta_{21} & 0 & \beta_{23} & \dots & \beta_{2n} \\ \vdots & \vdots & \vdots & \ddots & \vdots \\ \beta_{n1} & \beta_{n2} & \beta_{n3} & \dots & 0 \end{bmatrix}_{t=1} \begin{bmatrix} 0 & \beta_{12} & \beta_{13} & \dots & \beta_{1n} \\ \beta_{21} & 0 & \beta_{23} & \dots & \beta_{2n} \\ \vdots & \vdots & \vdots & \ddots & \vdots \\ \beta_{n1} & \beta_{n2} & \beta_{n3} & \dots & 0 \end{bmatrix}_{t=2} \dots \begin{bmatrix} 0 & \beta_{12} & \beta_{13} & \dots & \beta_{1n} \\ \beta_{21} & 0 & \beta_{23} & \dots & \beta_{2n} \\ \vdots & \vdots & \vdots & \ddots & \vdots \\ \beta_{n1} & \beta_{n2} & \beta_{n3} & \dots & 0 \end{bmatrix}_{t=n_t}$$

which represents the gross area changing from one land use to another that year. For example, $\beta_{23}$ is the area changing from land-use type 2 to land-use type 3 in km$^2$. The transition matrix at time $t$ can be derived from $\mathbf{U}_t$ by comparison with the previous layer $\mathbf{U}_{t-1}$. Each element is given by

$$\beta_{ijt} = \sum_{x=1}^{n_x} \sum_{y=1}^{n_y} [U_{xyt-1} = i \wedge U_{xyt} = j] l^2 \tag{3}$$

.

At each time step, the net change in the area occupied by each land use is given by the gross gains (the vector of column sums, $\mathbf{G}_t$) minus the gross losses (the vector of row sums, $\mathbf{L}_t$):

$$\Delta A_{ut} = G_{ut} - L_{ut} \tag{4}$$

where

$$G_{ut} = \sum_{i=1}^{n_u} \beta_{iut}$$

$$L_{ut} = \sum_{j=1}^{n_u} \beta_{ujt}$$

and $i$ and $j$ are the row and column indices.

We thus have three data structures, **U**, **B**, and **A**, which are inter-related by equations 1 - 4. **U** contains complete information about the system, which can be summarised in the form of **A** and **B**. **B** contains partial information about the system, which can be summarised in the form of **A**, but does not directly specify **U**. In itself, **A** does not directly specify either **U** or **B**, but can be used as a constraint in their estimation.

Multiple data sources are available which provide information in the form of these different data structures. Our approach here is to use equations 1 - 4 as a simple model to relate the different observational data via Bayesian data assimilation in a two-stage process. Firstly, we use a Bayesian approach to estimate the parameters in **B**, given prior information and partial observations of **U** and **A**. Secondly, we use the posterior distribution of **B** and spatial and probabilistic information on the location of land-use change to simulate posterior realisations of **U**. The maximum *a posteriori* probability (MAP, the mode of the posterior distribution) realisations represent our best estimate of land use and land-use change, given the available data.

## Data sources

We combined a number of data sources (Table 1) to describe the spatial and temporal change in land use in Scotland in the approach outlined above. A classification scheme was produced for each of these to aggregate the data into the broad classes used by Bradley *et al.* (2005 - forest, crop, grassland, rough grazing, urban, and other), close to the IPCC land-use

<sup>137</sup> classes (Penman et al., 2003). This was considered coarse enough that differences between <sup>138</sup> classifications could be aggregated into these six common classes, so that translation between <sup>139</sup> classifications did not cause major problems. In this classification, "grassland" comprises <sup>140</sup> all improved and actively managed agricultural grassland. "Rough grazing" comprises all <sup>141</sup> unmanaged grassland and semi-natural land. All spatial data were rasterised on a common <sup>142</sup> 100-m resolution grid, defined in the GB Ordnance Survey transverse Mercator projection. <sup>143</sup> The time domain considered was 1969 to 2015.

| Abbreviation | Data source | Data structures | Temporal coverage |
|---|---|---|---|
| CS | Countryside Survey | $\mathbf{B}$ | 1978, 1984, 1990, 2000, 2007 |
| AC | Agricultural Census | $\mathbf{A}$ | 1969-2016 |
| EAC | EDINA Agricultural Census | $\mathbf{G}$, $\mathbf{L}$, $\mathbf{W}$ | 1969-2016 |
| Corine | Corine | $\mathbf{U}$, $\mathbf{B}$, $\mathbf{W}$ | 1990, 2000, 2006, 2012 |
| IACS | Integrated Administration and Control System | $\mathbf{U}$, $\mathbf{B}$, $\mathbf{W}$ | 2004-2015 |
| NFEW | FC National Forest Estate and Woodlands | $\mathbf{U}$, $\mathbf{B}$, $\mathbf{W}$ | 1969-2014 |
| FC | FC new planting | $\mathbf{G}_{\text{forest}}$ | 1969-2016 |
| LCM | CEH Land Cover Map | $\mathbf{A}_{\text{urban}}$, $\mathbf{U}$, $\mathbf{W}$ | 1990, 2000, 2007, 2015 |
| ALCM | Agricultural Land Capability Map | $\mathbf{W}$ | NA |

Table 1: Data sources assimilated in the estimation of land-use change in Scotland.

## Data assimilation

<sup>145</sup> Our data assimilation method is represented graphically in 2 and proceeded as follows.

<sup>146</sup> • From repeat ground-based surveys, the CEH Countryside Survey (CS, Norton et al.,
<sup>147</sup>   2012; Wood et al., 2017) provides direct observations of $\mathbf{B}$ for approximately 150 1-km$^2$
<sup>148</sup>   survey squares in Scotland. Whilst the coverage is not large compared to the total area
<sup>149</sup>   of Scotland, the sample squares were chosen on a stratified design, and the observations
<sup>150</sup>   are valuable in having consistent recording methods over a long time period. The
<sup>151</sup>   method for scaling these survey squares to national scale is described in (Milne and

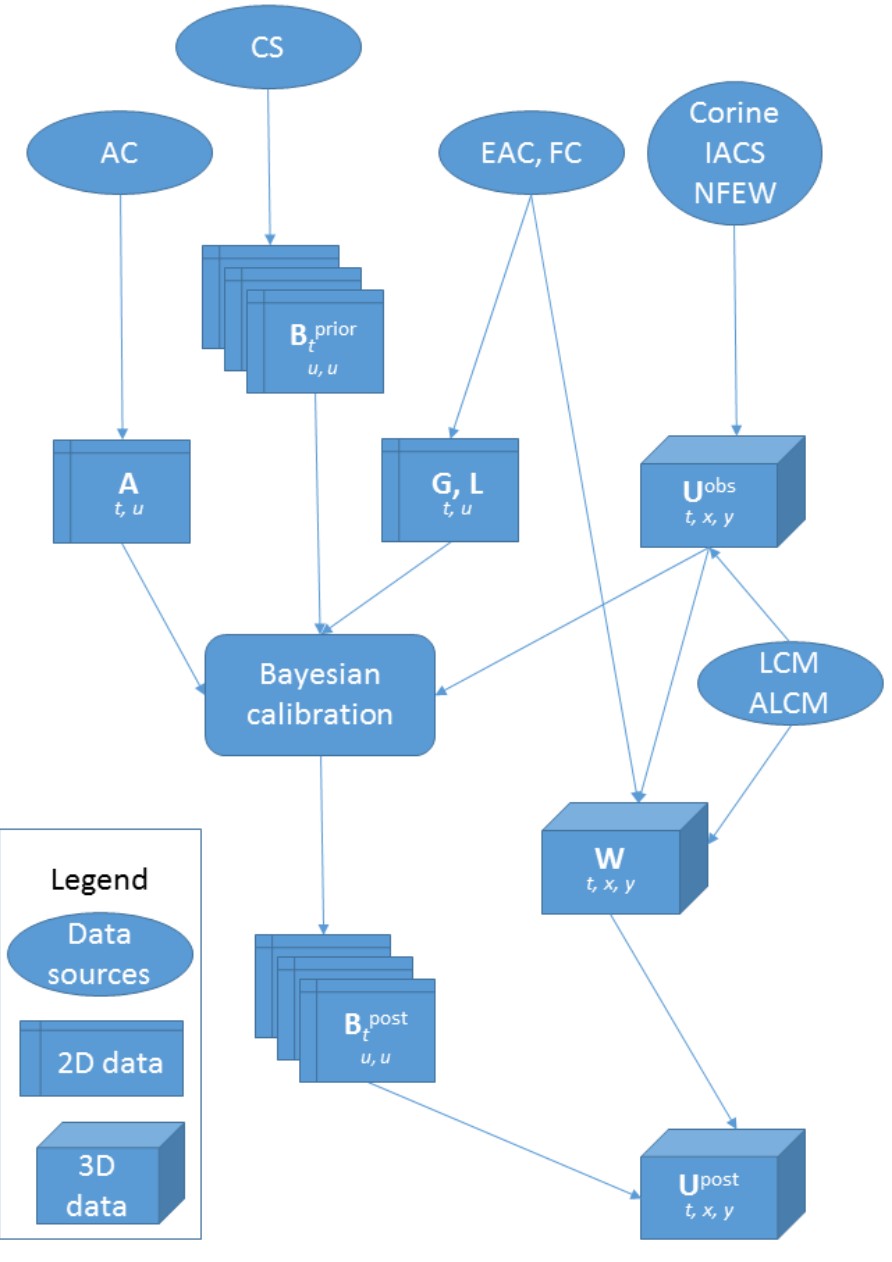

Figure 2: Schematic diagram showing information flow in the data assimilation procedure. Data sources are listed in Table 1. The prior estimate of the transition matrix $\mathbf{B}$ at each time point is provided by the CEH Countryside Survey (CS). Observations of the area ($\mathbf{A}$) occupied by each land use type $u$, the gross gains and losses($\mathbf{G}$ and $\mathbf{L}$), and spatially-explicit estimate of land use ($\mathbf{U}^{\text{obs}}$) are combined in a Bayesian calibration via the likelihood functions (equations 5 - 7) to produce updated, posterior estimates of the transition matrix $\mathbf{B}^{\text{post}}$. We then use spatial and probabilistic information on the location of land-use change ($\mathbf{W}$) to simulate posterior realisations of land use and land-use change ($\mathbf{U}^{\text{post}}$).

<sup></sup> Brown, 1997). Surveys were carried out in 1978, 1984, 1990, 2000, and 2007, and we interpolated linearly between survey years to produce an annual time series. We used the estimates derived in this way as our prior distribution of **B**. Each year, the mean of the prior distribution was taken to be the value of **B** from CS. The standard deviation $\sigma$ of the prior distribution was estimated from an earlier bootstrapping approach applied to the CS data (Scott, 2008), in an attempt to provide confidence intervals on the national-scale estimates of the areas of land-use transition (i.e. the **B** matrix).

- National Agricultural Census (AC) data provide annual records of the total area in the main agricultural land uses (Scottish Government, 2017). The Agricultural Census is conducted in June each year by the government agriculture department. Farmers declare the agricultural activity on their land in the form of ca. 150 items of data via a postal questionnaire. The results are collated at national scale. These are a long-running data set with near-complete coverage of agricultural land, relatively consistent over time, and are reported as national statistics and to the FAO. Hence it is desirable for our estimates of land-use change to be consistent with these data as far as possible. We therefore use these data as observations of $A_{ut}$ in the Bayesian framework, and predict $\Delta A_{ut}$ from $\mathbf{B}_t$ according to equation 4. The likelihood of the net change observed by Agricultural Census ($\Delta A_{ut}^{\mathrm{obs}}$) arising from normal distributions with means determined by equation 4 and the parameter matrix **B** is

$$\mathcal{L}_{\mathrm{net}} = \prod_{\substack{u=1 \\ t=1}}^{\substack{n_u \\ n_t}} \frac{1}{\sigma_{ut}^{\mathrm{obs}}\sqrt{2\pi}} \exp(-(\Delta A_{ut}^{\mathrm{obs}} - \Delta A_{ut}^{\mathrm{pred}})^2 / 2\sigma_{ut}^{\mathrm{obs}\,2}) \tag{5}$$

where $\Delta A_{ut}^{\mathrm{pred}}$ is the prediction from equation 4 for the change in land use $u$ at time $t$, and $\sigma_{ut}^{\mathrm{obs}}$ is the observational error in the Agricultural Census. So, we now have (i) a simple model which predicts net land-use change in terms of a parameter matrix; (ii) prior estimates of these parameters for each year from the Countryside Survey; and (iii) a function (equation 5)

<sup>175</sup> for the likelihood of the observations of net change given the model parameters. Combining

<sup>176</sup> these in Bayes Theorem, we can estimate the posterior distribution of the parameters, the

<sup>177</sup> transition matrix **B**. However before describing this, we can extend this simplest likelihood

<sup>178</sup> function by adding further sources of observational data.

<sup>179</sup> • The EDINA Agricultural Census (EAC) data (http://agcensus.edina.ac.uk/) provide

<sup>180</sup> additional information on land-use change, as they attempt to produce a spatially explicit

<sup>181</sup> version of the national-scale Agricultural Census data. Farm-level data is aggregated

<sup>182</sup> to 2-km grid cells, and data are available (or can be inferred) annually. While not

<sup>183</sup> containing explicit information on the actual land-use transitions, the resolution of the

<sup>184</sup> data is high enough that the net changes recorded each year in each 2-km cell may

<sup>185</sup> approximate the gross changes. In other words, because the data records the annual

<sup>186</sup> increases and decreases in land use across the grid of 2-km cells, the national totals of

<sup>187</sup> these increases and decreases gives an estimate of the gross change, the row and column

<sup>188</sup> sums of the transition matrix **B**, as well as the net change. When calculating the

<sup>189</sup> likelihood in our Bayesian framework, we can thus use the more informative observations

<sup>190</sup> of gross gains and losses (**G** and **L**) rather than just the observations of net change

<sup>191</sup> ($\Delta$**A**) from the national Agricultural Census. However, we know that the observations

<sup>192</sup> will tend to underestimate the gross change, because of the nature of the data reporting

<sup>193</sup> process: any counter-balancing gross change within the 2-km square is not included. To

<sup>194</sup> account for this, we can use a skewed normal distribution to represent this, such that

<sup>195</sup> predictions which overestimate the observations are more likely than underestimates.

<sup>196</sup> A skewed normal distribution of this form (Azzalini, 2017) gives the likelihood of the

<sup>197</sup> gross changes observed as:

$$\mathcal{L}_{\text{gross}} = \prod_{\substack{u=1 \\ t=1}}^{\substack{n_u \\ n_t}} \frac{2}{\sigma_{L_{ut}^{\text{obs}}}} \phi\left(\frac{L_{ut}^{\text{obs}} - L_{ut}^{\text{pred}}}{\sigma_{L_{ut}^{\text{obs}}}}\right) \Phi\left(\alpha\left(\frac{L_{ut}^{\text{obs}} - L_{ut}^{\text{pred}}}{\sigma_{L_{ut}^{\text{obs}}}}\right)\right)$$

$$\times \frac{2}{\sigma_{G_{ut}^{\text{obs}}}} \phi\left(\frac{G_{ut}^{\text{obs}} - G_{ut}^{\text{pred}}}{\sigma_{G_{ut}^{\text{obs}}}}\right) \Phi\left(\alpha\left(\frac{G_{ut}^{\text{obs}} - G_{ut}^{\text{pred}}}{\sigma_{G_{ut}^{\text{obs}}}}\right)\right) \tag{6}$$

where $\phi$ is the standard normal probability density function, $\Phi$ is the corresponding cumulative density function, and $\alpha$ is the skew parameter. Positive $\alpha$ produces a positive skew (when $\alpha = 0$ we have the standard normal distribution). The parameter $\alpha$ can itself be estimated as part of the data assimilation procedure.

- Several data sources provide observations of $\mathbf{U}$ for one or more land uses at a restricted set of time points. We combine these into a single array $\mathbf{U}^{\text{obs}}$ as follows.

  – For an initial estimate of $\mathbf{U}$, we use the Corine data sets for 1990, 2000, 2007, and 2012 (European Environment Agency, 2016). For each grid cell, change between these years was assumed to occur at a random time within the interval, so that at national scale we effectively interpolate linearly. This produces $\mathbf{U}$ with complete UK coverage at annual resolution over the period 1990 to 2012.

  – We overlay this with IACS data over the period 2004 to 2015 (Tomlinson et al., 2017). The Integrated Administration and Control System (IACS) is a European-wide spatially explicit dataset at the field level that serves as a register of agricultural subsidy claims under the EU Common Agricultural Policy. IACS records field-level land use (crop type, grassland age, forest coverage), field geometry and its association to a farm holding. This has large, but not complete spatial coverage (65 % of the Scottish land area), and the Corine data are retained where IACS data are missing. Where there are conflicts with Corine, IACS data are given precedence because they are direct ground-based records.

  – We then add forestry data from the GB Forestry Commission (FC) National

<sup>219</sup> Forest Estate and Woodlands (https://www.forestry.gov.uk/datadownload), which

<sup>220</sup> records the location and planting date of forestry. Again, this only has limited

<sup>221</sup> coverage, as it only covers forest land, but is given precedence in the case of conflict

<sup>222</sup> with the Corine/IACS data. We iterate over each time step to calculate $\mathbf{B}_t^{\mathrm{obs}}$ with

<sup>223</sup> equation 3. $\mathbf{B}_t^{\mathrm{obs}}$ thus contains an observed estimate of the transition matrix for

<sup>224</sup> each year, from the combination of Corine, IACS and FC data.

<sup>225</sup> We can therefore add an additional term to the likelihood function which incorporates the

<sup>226</sup> comparison of the observations $\mathbf{B}^{\mathrm{obs}}$ with the values in the current parameter set $\mathbf{B}^{\mathrm{pred}}$.

<sup>227</sup>

$$\mathcal{L}_{\mathbf{B}} = \prod_{\substack{i=1 \\ j=1 \\ t=1}}^{\substack{n_u \\ n_t}} \frac{1}{\sigma_{\beta_{ijt}^{\mathrm{obs}}}\sqrt{2\pi}} \exp(-(\beta_{ijt}^{\mathrm{obs}} - \beta_{ijt}^{\mathrm{pred}})^2 / 2\sigma_{\beta_{ijt}^{\mathrm{obs}}}^2) \tag{7}$$

<sup>228</sup> • To establish the posterior distribution, we use the Markov Chain Monte Carlo (MCMC)

<sup>229</sup> approach with the "DEz" algorithm implemented in the R package `BayesianTools`

<sup>230</sup> (Hartig et al., 2017). For each interval in the 46 year time series, an MCMC simulation

<sup>231</sup> was run, using the prior $\mathbf{B}_t$ matrix from Countryside Survey, the observations of $\Delta\mathbf{A}_t$,

<sup>232</sup> $\mathbf{L}_t$, $\mathbf{G}_t$ for that year, and the observed $\mathbf{B}_t$ matrix from Corine-IACS_NFEW. In practice,

<sup>233</sup> it is more convenient to use log-likelihoods, and our overall likelihood was the summation

<sup>234</sup> of $\log(\mathcal{L}_{\mathrm{net}})$, $\log(\mathcal{L}_{\mathrm{gross}})$ and $\log(\mathcal{L}_{\mathbf{B}})$. Nine chains were used, with 100,000 interations in

<sup>235</sup> each. To establish the initial $\mathbf{B}$ parameter values for one of the chains, a least-squares fit

<sup>236</sup> with the $\Delta\mathbf{A}$ was used. Other chains were over-dispersed by adding random variation

<sup>237</sup> to this best-fit parameter set.

<sup>238</sup> • Having established the posterior distribution of $\mathbf{B}$, we use spatial and probabilistic

<sup>239</sup> information on the location of land-use change to simulate posterior realisations of

<sup>240</sup> $\mathbf{U}^{\mathrm{post}}$. Starting with our best estimate of the near-present state of land use, $\mathbf{U}_{t=2015}^{\mathrm{obs}}$,

<sup>241</sup> we work backwards in time. At each time step, we know the number of grid cells which

need to change from land use $i$ to land use $j$ from the posterior matrix $\mathbf{B}_t$. For each $i$ to $j$ transition, we perform a weighted sampling operation to select this number of cells from those where $U_{xyt} = i$. In choosing which cells to assign to $j$, we use the available data to calculate the probabilities which weight the sampling. Recall that $\mathbf{U}^{\text{obs}}$ is given by the amalgamation of Corine, IACS and NFEW data. In the simplest case, the probabilities are determined only by this: all cells where $U_{xyt}^{\text{obs}} = i$ and $U_{xy,t-1}^{\text{obs}} = j$ have equally high probability of being selected in the sample, and all cells where $U_{xyt}^{\text{obs}} = i$ and $U_{xy,t-1}^{\text{obs}} \neq j$ have equally low (but non-zero) probability of being selected in the sample. This requires only a few simple rules to construct the probability weightings, $\mathbf{W}$, for sampling cells for conversion from $i$ to $j$:

$$\text{if } U_{xy,t}^{\text{obs}} \neq i \text{ then } W_{xy} \leftarrow 0 \text{ else } W_{xy} \leftarrow 1$$
$$\wedge \quad \text{if } U_{xy,t-1}^{\text{obs}} = j \text{ then } W_{xy} \leftarrow 1 \text{ else } W_{xy} \leftarrow p_m$$

where $p_m$ is the probability of cells being misclassified in $\mathbf{U}^{\text{obs}}$, which we estimate to be 0.05. Sampling is done without replacement, so that a grid cell can only be selected once per year. To illustrate with an example, we start with our current map of land use, $\mathbf{U}_{t=2015}^{\text{obs}}$. Suppose our posterior estimate of $\mathbf{B}_t$ determines that seven grid cells change from crop to grass, as we go back to 2014.Only cells which are crop in 2015 are valid candidates. Of these, those which were grass in 2014 (according to $\mathbf{U}^{\text{obs}}$) will have high probability of being selected; others will have a low probability. If the posterior $\beta_{ijt}^{\text{post}}$ area is lower than $\beta_{ijt}^{\text{obs}}$, not all the cells with high weightings from the above rules will be selected in the sample. If the posterior $\beta_{ijt}^{\text{post}}$ area is higher than $\beta_{ijt}^{\text{obs}}$, additional cells, with low weightings from the above rules, will be selected in the sample. Thus, the cells which we are likely to change are those which are designated by $\mathbf{U}^{\text{obs}}$ as crop in 2015 and grass in 2014. The effect of this is to generally recreate the spatial and

temporal pattern seen in $\mathbf{U}^{\text{obs}}$ (data from Corine, IACS and NFEW), but modified according to the extent of change estimated in the posterior $\mathbf{B}^{\text{post}}$.

- As well as using the data from Corine, IACS and NFEW, we can also use other spatial data sets to inform the location of land-use change in our simulatations of the posterior $U_{xyt}$. Any spatial data set which gives information on where and when a land use or land-use change occurs can be incorporated into the weighting used for sampling. Here, we used three additional data sets.

    - EDINA Agricultural Census gives an estimate of $\Delta\mathbf{A}$ at 2-km resolution. For each land use, an observed increase in area indicates the likely location of predicted gains. We therefore add a term to $\mathbf{W}$ which is proportional to $\Delta\mathbf{A}$.

    - The CEH Land Cover Map (Rowland et al., 2017) gives an estimate of $\mathbf{U}_t$ in 1990, 2000, 2007, and 2015 at high spatial resolution. Occurrence of a land use in the LCM suggests an area where gains would be more likely to occur. We add a term to $\mathbf{W}$, based on occurrence of that land use in the LCM.

    - Agricultural Land Capability Maps gives an estimate of how suitable land is for intensive agriculture, with a scale which ranges from good arable land, through intensive grassland and extensive grassland, to rough grazing. This scale can be translated into a probability of occurence for the land uses considered here, and added into the weighting of the sampling again. We use all the above information to produce many posterior realisations of $\mathbf{U}^{\text{post}}$, using the posterior $B$ matrix and the sampling process described earlier.

Because the $\mathbf{U}$ data structure is large, we are limited in simulating many samples. It is therefore useful to summarise as the much smaller set of unique vectors and their corresponding areas. Our approach is to simulate 1000 samples, to calculate the unique vectors and their areas, and not to retain the larger data structure to reduce storage requirements. Another possible approach would be to simulate using only the MAP $B$ matrix, and thereby generate the most likely realisations of $U_{xyt}$, rather than the whole posterior distribution.

## Carbon dynamics following land use change

We applied a simple empirical model of carbon fluxes following land use change, based on the UK LULUCF GHG inventory (Griffin et al., 2014). The soil component is based on the work of Bradley et al. (2005), and uses an analysis of the total soil carbon stock in a large number of soil cores, classified by land use and soil series. A linear mixed-effects model was applied to these data, to quantify the average effect of land use on soil carbon stock, treating soil series as a random effect. The model uses these mean values to represent the equilibrium soil carbon stock for each land-use class. When land use changes, the soil carbon stock moves towards the equilibrium soil carbon stock for the new land use. The soil carbon stock at location (x,y) and time $t$ is given by:

$$C_{xyt} = C_u^{\text{eq}} - (C_u^{\text{eq}} - C_{xy,t-1}) \exp(-k\Delta t) \tag{8}$$

where $C_u^{\text{eq}}$ is the equilibrium soil carbon stock for the current land use $u$, $C_{xy,t-1}$ is the soil carbon stock at the previous time step, and $k$ is a rate constant. The flux of carbon over the time step, $\Delta t$, is given simply by difference:

$$F_C = C_{xyt} - C_{xy,t-1} \tag{9}$$

The above-ground component applies to the growth of biomass following afforestation, and uses the yield tables for British forestry produced by Edwards & Christie (1981), as interpolated and expanded to include non-merchantable timber biomass and wood products by Dewar & Cannell (1992). The mean change in above-ground biomass was assumed to be negligible in other land-use transitions in this simple model.

# Results

Because of the availability of remotely-sensed data products, we are relatively confident in the present-day distribution of land use (Figure 3). This shows the concentration of urban areas in Scotland in the central belt, the restriction of cropland to the drier, flatter east coast, improved grassland mainly in the lowlands in the wetter south and west, and rough grazing and forestry sharing the Southern Uplands and Highlands in the north and west.

As an initial step in the data assimilationn process, a close least-squares fit to $\Delta \mathbf{A}$ was achieved within a few tens of iterations, indicating that there were no particular numerical difficulties in estimating the $\mathbf{B}$ parameters. Standard measures were applied to assess whether the posterior distribution of $\mathbf{B}$ was suitably characterised by the output of the MCMC sampling. As well as inspection of the trace plots and the form of the distribution of the $\mathbf{B}$ parameters, we calculated the effective sample size, the acceptance rate, and various standard convergence diagnostics (Gelman and Rubin, 1992; Geweke, 1992; Raftery and Lewis, 1992). All of these showed satisfactory performance, that the MCMC chains converged, and that nine chains with 100,000 samples provides a reasonable estimate of the posterior distribution of $\mathbf{B}$.

Figure 4 shows the Agricultural Census observations, and posterior predictions of the net change in area of each land-use class. The net change implied by the prior CS and IACS observations of $\mathbf{B}$ are also shown. The broad trends are: (i) an increase in forest cover due to sustained commercial forest planting; (ii) a corresponding decrease in rough grazing and semi-natural land due to expansion of forestry and improved grassland; (iii) an increase in cropland area between 1970 and 1990, with subsequent decline to the present day, due to changes in economic forces and subsidy incentives; (iv) an increase in grassland area since around 1990, partly corresponding to the reduction in crop area, and partly due to a general expansion on to rough grazing areas; and (v) a slow but consistent expansion of the urban area. These trends are picked up by the different sources of observations to some extent. The

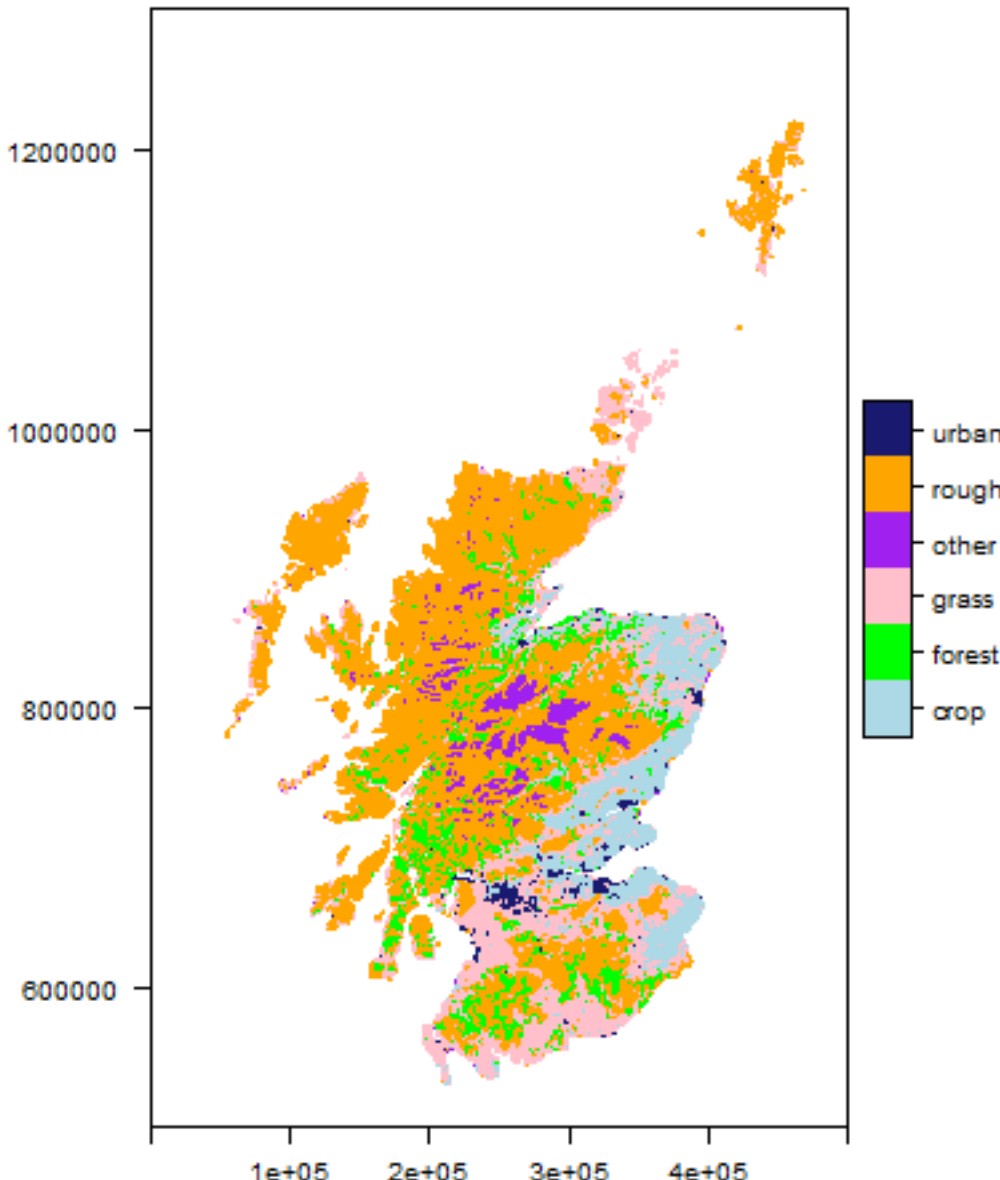

Figure 3: Land use in Scotland in 2015 as estimated by the CEH Land Cover Map. "Grass" comprises all improved and actively managed agricultural grassland. "Rough" includes all rough grazing, unmanaged grassland and semi-natural land. "Other" comprises barren areas such as montane and coastal areas. Map coordinates are in British National Grid. For legibility, we show data aggregated to 2-km squares, though they are available at 25-m resolution.

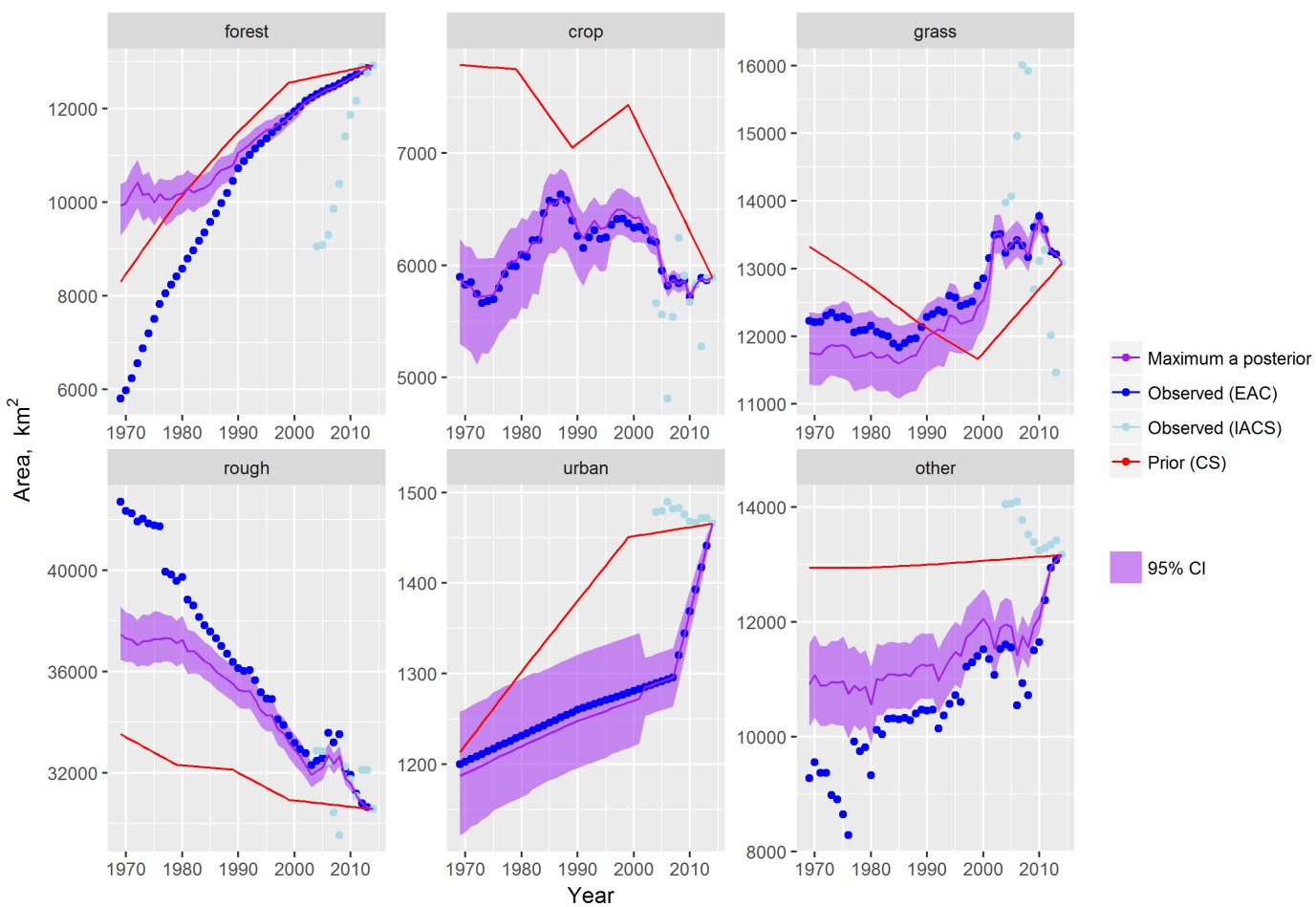

Figure 4: Time series of the area occupied by each land use ($A_{ut}$) from 1969 to 2015, showing the observations, prior and posterior estimates. The shaded band shows the 2.5 and 97.5 % percentiles of the posterior distribution of the net change in area.

Agricultural Census has near-complete coverage, and annual resolution, so shows a detailed pattern, to which we give most credence. The CS data, used as the prior, have only decadal time resolution, but pick up these general trends, and approximate the same pattern as seen in the Agricultural Census data. The IACS data show considerable year-to-year variability, and tend to show exaggerated net changes compared to AC. The posterior prediction generally falls in between the AC observations and the CS prior, but tracks closer to the AC.

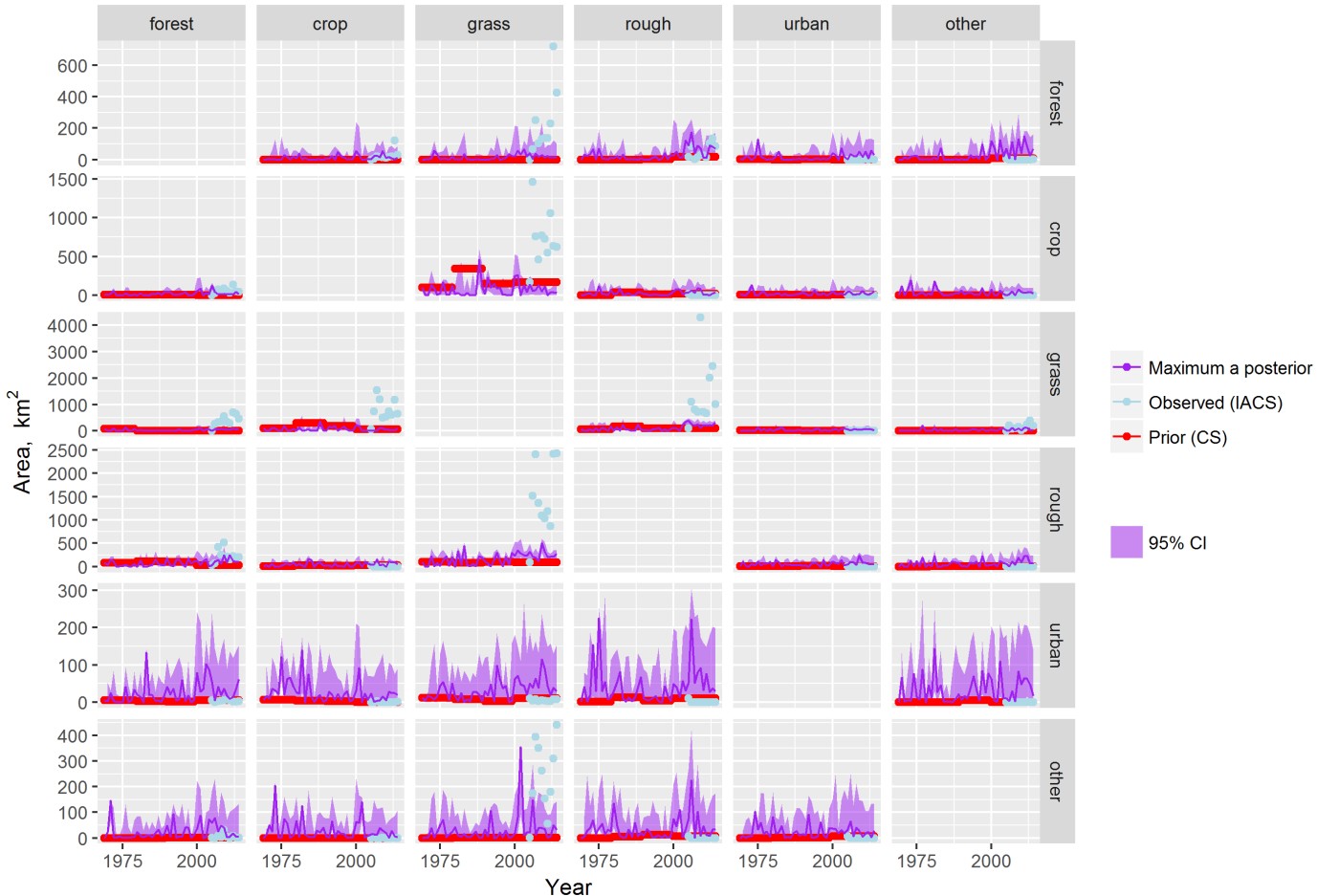

Figure 5: Prior and posterior distributions of the transition matrix **B**, representing the gross area changing from the land use in each row $i$ to the land use in each column $j$ each year from 1969 to 2015. Red lines show the prior estimate from the Countryside Surveys. Pale blue points show estimates from IACS plus Corine and NFEW. The maximum *a posteriori* estimates after assimilating all data sources are shown in purple. The shaded band shows the 2.5 and 97.5 % quantiles of the posterior distribution. Note the y scale is different for each row.

CS provided our prior estimate of **B**. Given the relatively small spatial coverage of CS,

 uncertainty ($\sigma$) in the prior $\mathbf{B}$ is rather high. This would be expected to effectively limit the

influence of the prior on the posterior $\mathbf{B}$, compared to the observations from IACS, which

have national coverage. Figure 5 shows that estimates of $\mathbf{B}$ from these two data sources are

quite different. Particularly in the transitions to and from grassland, values of $\mathbf{B}$ from IACS

tend to be an order of magnitude larger than values from CS, and more variable. However,

the posterior $\mathbf{B}$ remains closer to the prior than might be expected. This is because values of

$\mathbf{B}$ close to the IACS observations are deemed unlikely with respect to the other terms in the

likelihood function. That is, the gross and net changes in area implied by the IACS data are

inconsistent with the other observations of $\mathbf{G}$, $\mathbf{L}$ and $\mathbf{\Delta A}$ from AC (Figures 4 - 7).

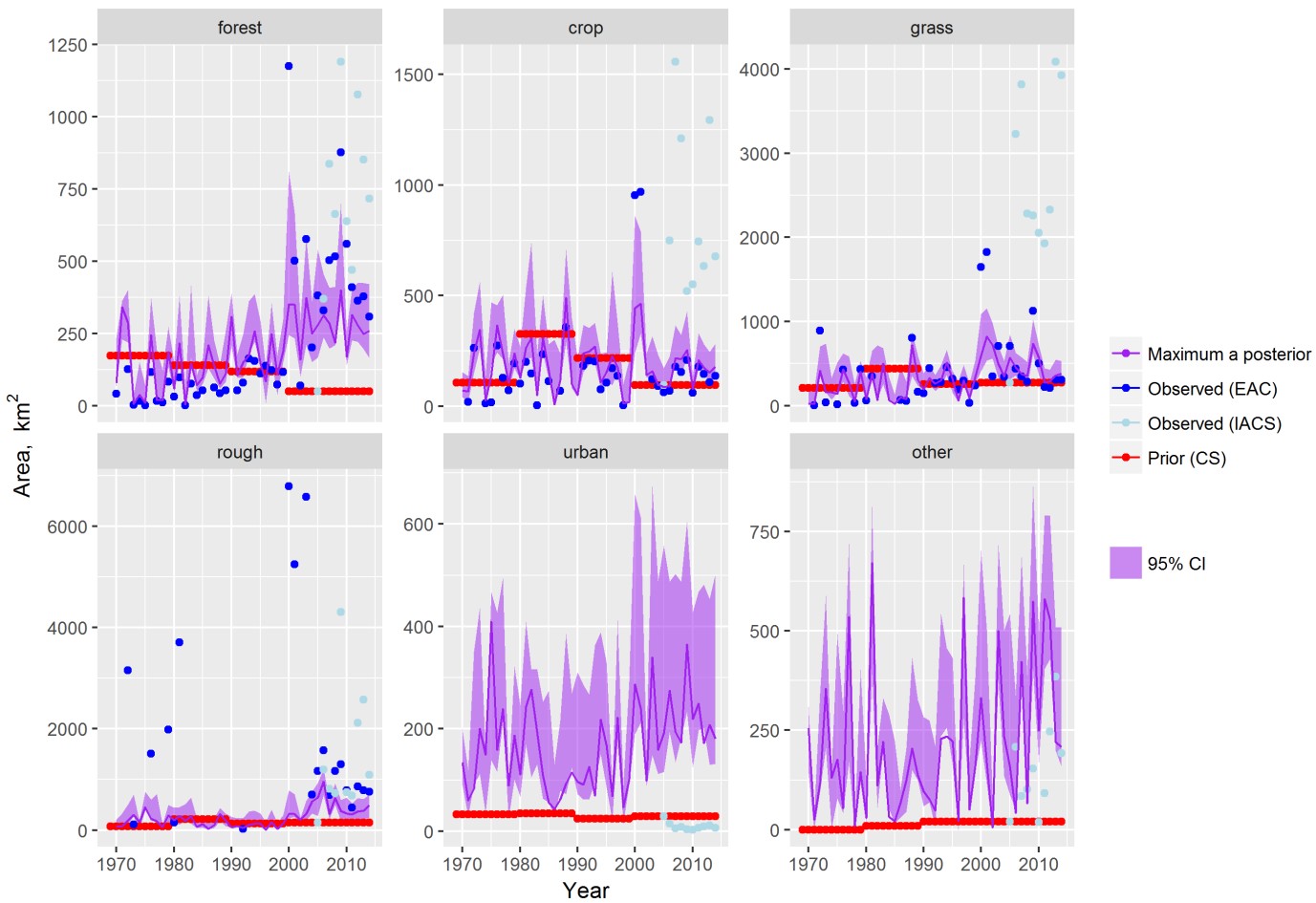

Figure 6: Time series of the gross gain in area of each land use ($A_{ut}$) from 1969 to 2015, showing the observations, prior and posterior estimates. The shaded band shows the 2.5 and 97.5 % percentiles of the posterior distribution.

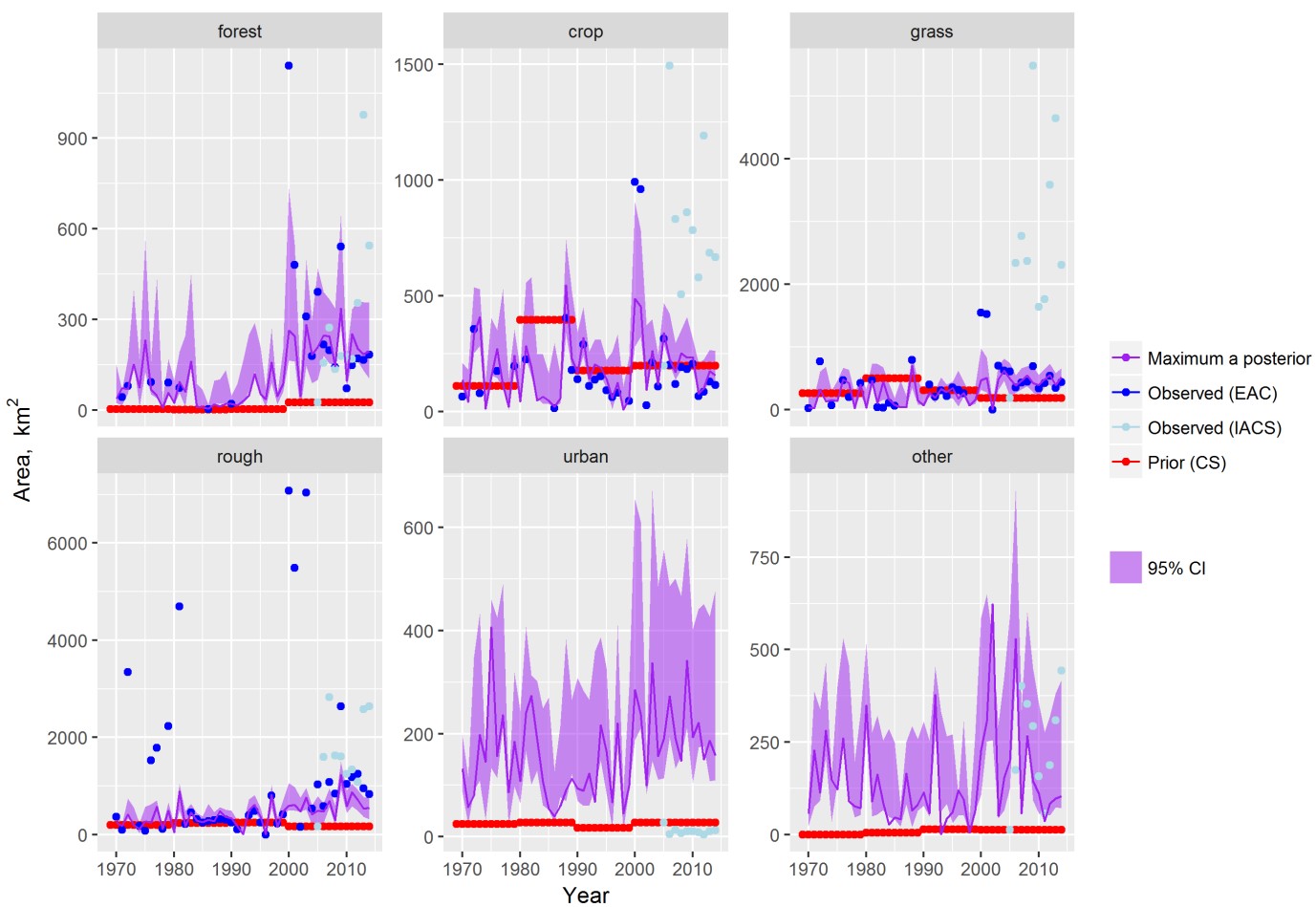

Figure 7: Time series of the gross loss in area from each land use $(A_{ut})$ from 1969 to 2015, showing the observations, prior and posterior estimates. The shaded band shows the 2.5 and 97.5 % percentiles of the posterior distribution.

For cropland and improved grassland, CS and EAC show general agreement on the magnitude and pattern in area gained and lost to each land use (Figure 6 and Figure 7). An exception is an apparent anomaly in the early 2000s, when EAC gains and losses are both around 1000 km$^2$ higher than average for two years. This is not reflected in the net changes reported in the AC, so has to be treated with some caution. Reported gains and losses of rough grazing are much higher and very variable in EAC. This variability does not seem closely linked to the net change reported at national scale, so again, we treat this with some scepticism. There are no data on the gross gains and losses of urban and other land-use areas, as they are not covered by the AC or CS, and these terms are less well constrained.

Figures 4 - 7 show that there is considerable spread in the posterior distribution of **B** and predictions of $\Delta$**A**. The 95 % credibility interval is typically of the order of 100 km$^2$ for the individual B parameters, and several hundred km$^2$ for the predictions of $\Delta$**A**. The credibility intervals are smallest where multiple data sources agree on the nature of land-use change, and where the change is coherent across land uses. That is, an increase in one land use has to be balanced by a decrease in one or more other land uses. We have less confidence in predictions where the observed change in one land use is not compensated for by other land use changes. Credibility intervals in $\Delta$**A** increase as we go back in time, because the uncertainty accumulates from year to year, although the increase has square root form rather than linear,

Figure 8 and Figure 9 attempt to convey the detailed structure of the posterior **U** in a simple graphical summary. Figure 8 shows the 100 most frequent vectors of land-use change. Line thickness and opacity are proportional to the frequency (= area) of each vector, so that the dominant vectors are the most visually obvious. The plot shows that a wide range of land-use transitions occurs over the time period considered. Transitions from rough grazing to forest and to improved grassland are dominant. Bi-directional transitions between crop and improved grassland are particularly common in the 1980s. This comes from information

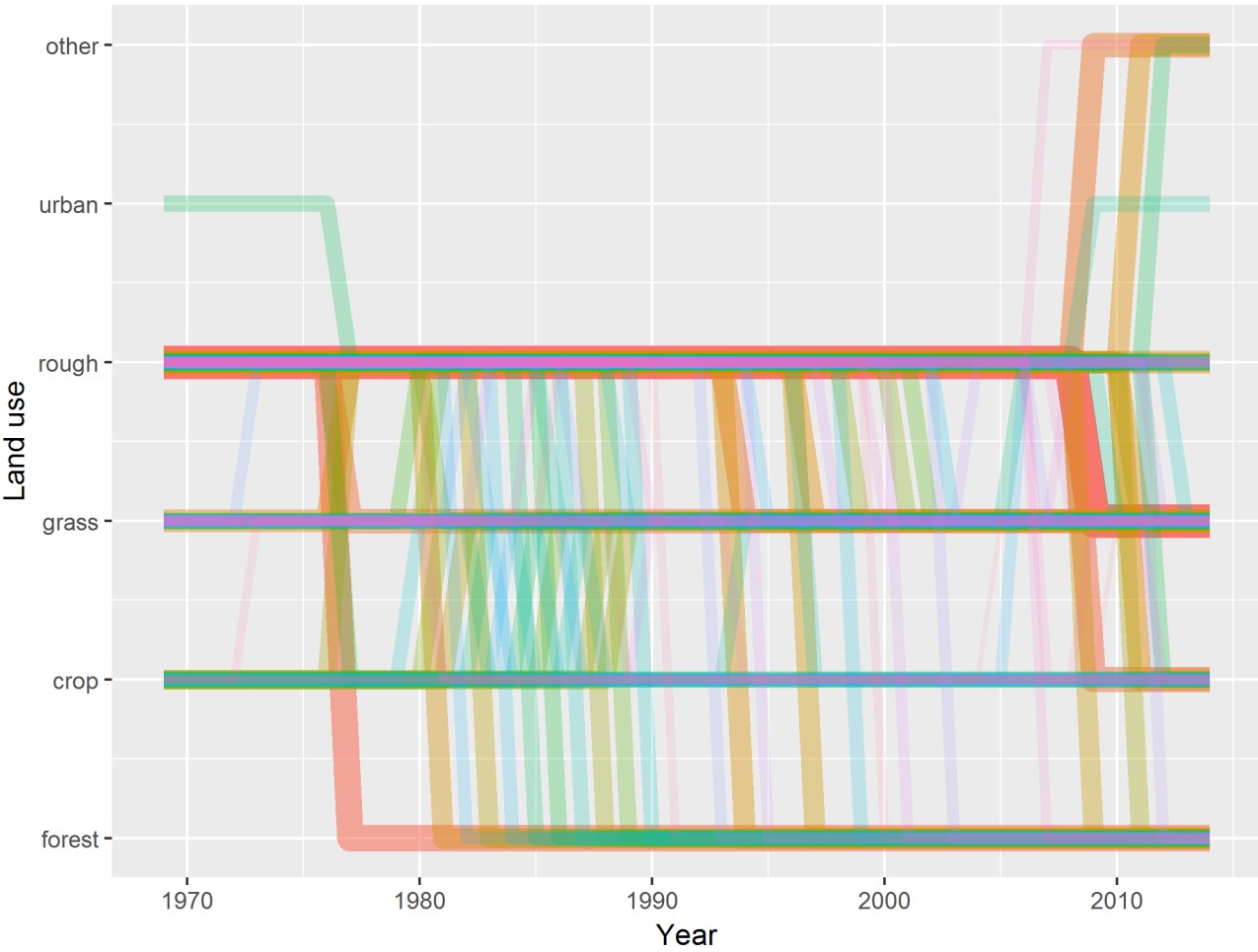

Figure 8: Trajectories of the 100 land-use vectors in the posterior $U$ with the largest areas (excluding the six vectors which show no change). Each vector of land use is shown in a different colour, varied arbitrarily to differentiate different vectors. Line thickness and opacity are proportional to the frequency of (or total area occupied by) each vector, so that the dominant vectors are the most visually obvious.

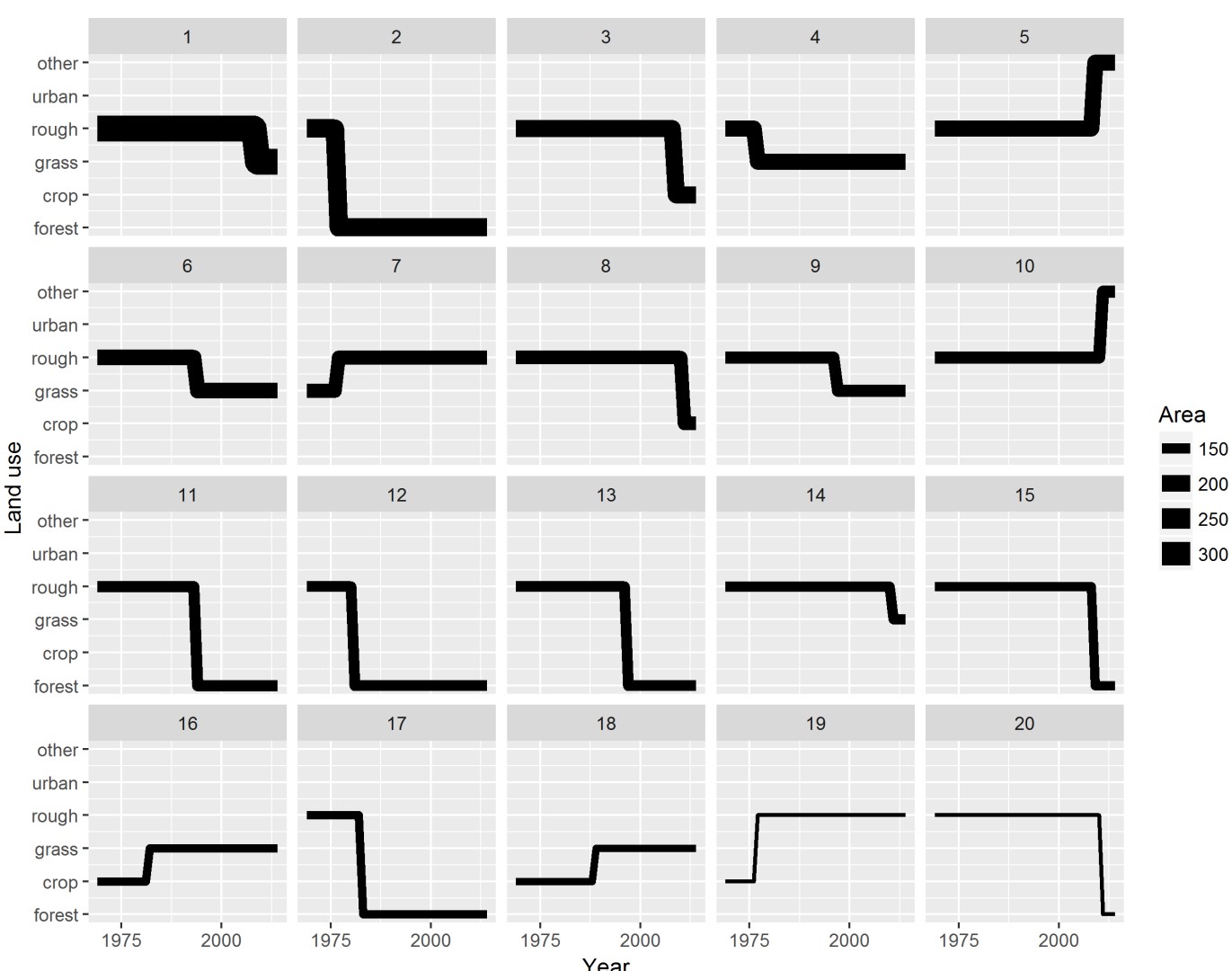

Figure 9: Trajectories of the 20 land-use vectors in the posterior $U$ with the largest areas (excluding the six vectors which show no change). Line thickness is proportional to the frequency of (or total area occupied by) the vector

in the prior, the **B** matrices from CS which shows markedly higher crop to grass and grass to crop conversion rates over this time.

Figure 9 shows the 20 most frequent vectors more clearly, with each vector on a separate panel. This shows that 17 out of 20 involve transitions to or from rough grazing (which includes all semi-natural) land, which is the largest land use in Scotland by some way (around half the total area). Seven of these represent afforestation, which has mainly occurred on less productive, upland rough grazing land. Five vectors represent expansion of improved grassland on to rough grazing land. Vectors with two or more changes are less frequent, with none occurring in the top 20, but do represent a significant part of the total area (~8 % of the area undergoing change).

Figure 10 shows the $CO_2$ flux resulting from land-use change over the 46-year period, derived from equations 8 - 9 and the posterior distribution of **U**. The positive fluxes denote a gain to the terrestrial carbon stock, negative fluxes represent a loss to the atmosphere. We only represent land-use change from 1969 onwards here, but the effects on carbon flux are long-lasting. Hence, the carbon flux calculated here is initially small, and increases as the area having undergone land-use change accumulates over time. The accumulation of carbon in forest biomass (and wood products) following afforestation over this period is the largest term in these results. The forest planting rate has decreased markedly since 2005, giving the reduction in carbon sequestration in recent years. In this simple soil model, land uses with higher equilibrium soil carbon than the average will tend to act as carbon sinks; those lower than the average will be sources. Carbon emissions from cropland increase as predominantly grassland is converted to cropland between 1970 and 1990. This then levels off as the cropland area remains stable or declines thereafter. Transitions to forest and rough grazing result in carbon sinks because they both have higher than average equilibrium soil carbon, and both show sizeable gross gains over the period. Rough grazing land also shows substantially larger gross area losses, but the associated carbon fluxes associated with this are attributed mainly

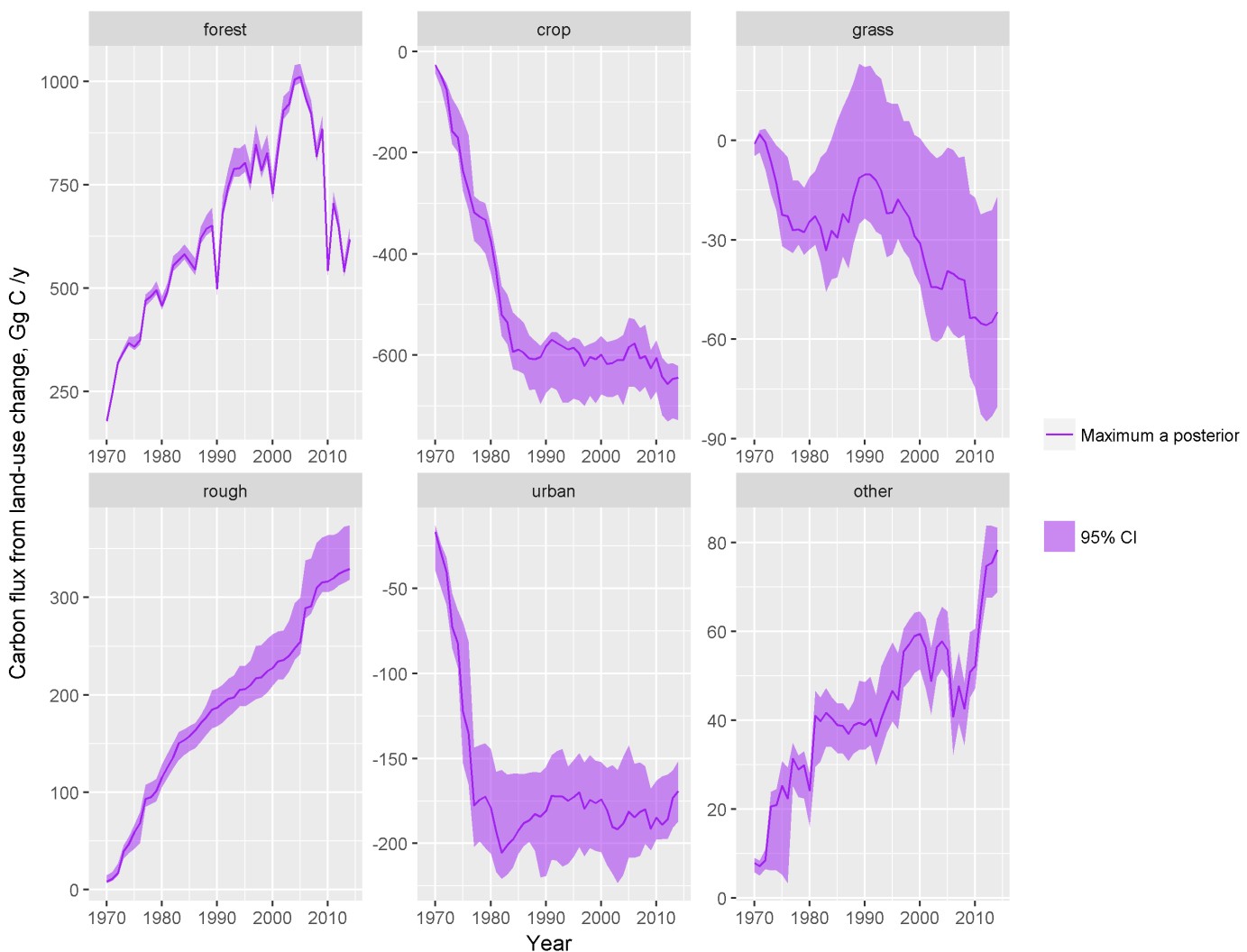

Figure 10: Net carbon flux from land-use change in Scotland over 1969-2015 showing the maximum *a posteriori* estimate and its 95 % credibility interval. The flux is attributed to change *to* each land-use class *u*. Positive fluxes denote a gain to the terrestrial carbon stock; negative fluxes represent a loss to the atmosphere.

to improved grassland, as this is the main land use to which it changes. Improved grassland therefore shows as a small net source of carbon, the result of land use changes from cropland to improved grassland (sink) and rough grazing to improved grassland (source).

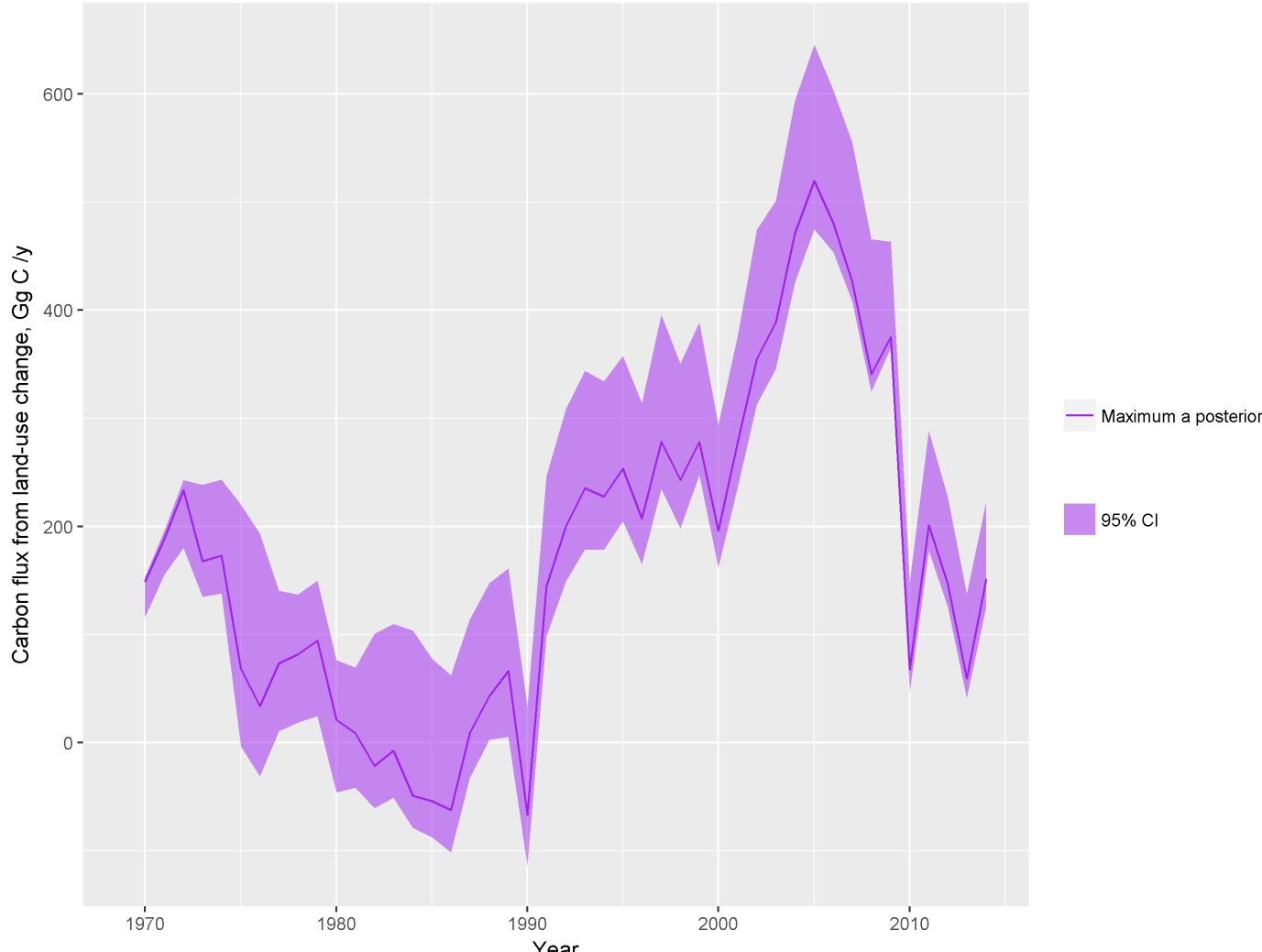

Figure 11: Total net carbon flux from land-use change in Scotland over 1969-2015, showing the maximum *a posteriori* estimate and the 95 % credibility interval. Positive fluxes denote a gain to the terrestrial carbon stock; negative fluxes represent a loss to the atmosphere.

The overall effect of these component fluxes is to produce a net sequestration of carbon from land-use change (Figure 11). The 95 % credibility interval in the near-present-day carbon flux is around 100 Gg C $y^{-1}$, close to 50 % of the best estimate. There is therefore considerable uncertainty in the carbon flux associated with land-use change, because the

underlying changes in land use are themselves uncertain. Recognition and propagation of this uncertainty is therefore important.

Mapping the carbon fluxes calculated by equations 8 - 9 and the MAP estimate of $\mathbf{U}$, we can see that the carbon fluxes closely follow the present-day land-use distribution (Figure 12). The carbon sinks are associated mainly with new forest areas, and to a lesser extent, wherever improved grassland or cropland has reverted to rough grazing. The carbon sources are associated with wherever cropland or urban areas have expanded.

# Discussion

The results show that we can provide improved estimates of past land-use change using multiple data sources in the Bayesian framework. The computation involved is quite feasible on a modern computer, requiring around three hours to estimate the parameters for a 46-year period. The output of the assimilation procedure provides vectors of land-use change in the form required for dynamic and process-based modelling, which we illustrate with the soil carbon modelling example. The main advantage of the approach is that it provides a coherent, generalised framework for combining multiple disparate sources of data.

As far as we are aware, there are no previous applications of formal data assimilation approaches to land-use change. However, some studies have addressed the same problem with related methods. Hurrt et al. (2011, 2006) used estimates of $\mathbf{A}$ together with estimates of wood harvest to predict $\mathbf{B}$. The study was carried out at global scale at 0.5 degree resolution, and covered both historical and future scenarios for the period 1500-2100. To make the problem tractable, the transition matrix $\mathbf{B}$ was initially specified for only three land uses, so that a unique minimum solution could be found. Additional transitions associated with shifting cultivation and wood harvest were then calculated in a further step. They used a rule-based model which specified assumptions about the residence time of agricultural land,

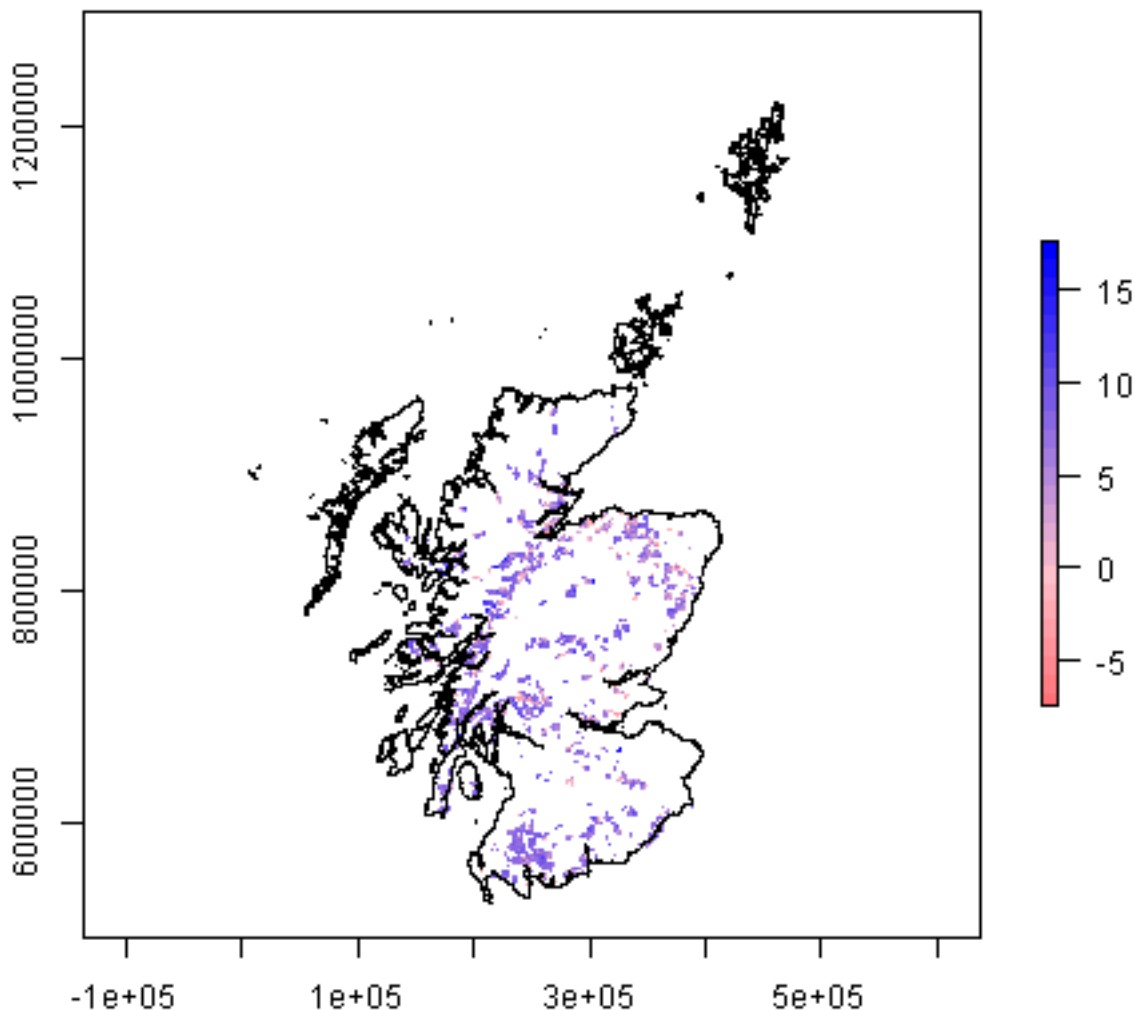

Figure 12: Net carbon flux (in kg C m$^{-2}$) from land use change in Scotland over 1969-2015 from the maximum *a posteriori* estimate of **U**. Positive fluxes denote a gain to the terrestrial carbon stock; negative fluxes represent a loss to the atmosphere. Map coordinates are in British National Grid.

the priority of land for conversion to agriculture and for wood harvesting, and the spatial pattern of wood harvesting within a country. The distribution of land use over space and time $\mathbf{U}$ was not explicitly represented; instead, the area and age of "secondary" land in each grid cell was tracked in a book-keeping approach. However, because only a matrix is calculated at each time step, the approach does not produce explicit vectors of land use for dynamic modelling, and such things as rotational land use are not easily represented. Sensitivity to various assumptions was analysed, but the uncertainties associated with the input data and these model assumptions cannot readily be quantified.

Fuchs et al. (2013) used a number of data sets, including that of Hurrt et al. (2006), to explicitly estimate the change in land use over space and time $\mathbf{U}$ for the whole of Europe at 1 km$^2$ resolution for each decade 1900-2010. Using logistic regression, they calculated "probability maps" for each land cover class, based on biogeophysical and socio-economic properties of each grid cell as explanatory variables for land use in 2000. For each decade and each country within the EU27, the net increase in the area of each land use (positive $\Delta A_{ut}$) was allocated to the grid cells with the highest probability score for that land use. This approach yields essentially the same data structure as our method, and is wider in scope, covering all of Europe.

Our method represents an advance on this in several ways. Because the approach of Fuchs et al. (2013) is based on net change in areas at country scale, the extent of the true, gross changes will be under-estimated, possibly by orders of magnitude, and implicitly the $\mathbf{B}$ matrices are minimised. Our approach uses explicit observations of the annual transition matrices $\mathbf{B}$ as far as possible. Rather than regression relationships, our approach uses annual spatially explicit observations of where and when land-use change is likely to have occurred (based on CS, IACS and EAC). We use higher temporal and spatial resolution (annually, at 100 m) because this is possible with the data available in the UK, and with the limited spatial domain we attempt to cover. At continental and global scales, the same quantity and

resolution of data is not available, and the computation issues become much larger. Our approach explicitly incorporates and propagates the uncertainty in the posterior distribution of **B** and predictions of **A** and subsequently modelled carbon fluxes. The uncertainty in land-use change is substantial, even in the UK where land management records are good. Our methodology accounts for this uncertainty in a mathematically rigorous way (Van Oijen, 2017), and propagates this through to the subsequent modelling of other outputs, such as soil carbon fluxes. On a fundamental level, the Bayesian approach gives the correct theoretical answer to the data assimilation problem: if the observational error and prior are correctly specified and the posterior is adequately characterised by the MCMC sampling, then the posterior correctly represents the actual state of knowledge about the system parameters and predictions (Gelman et al., 2013; Reich, 2015).

We thus need to consider how well we can characterise the observational error, and the prior and posterior distributions. Establishing that the posterior distribution has been adequately characterised by the MCMC sampling is relatively straightforward. There are various criteria for assessing this (the effective sample size, and measures of MCMC chain convergence) which the results meet. In this study we chose to use an informative prior based on CS. This follows the way in which the data became available chronologically; these were the only data available with which we could estimate land-use change in the UK when an inventory of carbon emissions was first attempted (Cannell et al., 1999). The uncertainty in the prior distribution of **B** can be relatively well quantified, because considerable effort has gone into quantifying the likely level of error in the national-scale estimates of land use (Scott, 2008; Wood et al., 2017). The standard deviation $\sigma$ of the prior distribution was most easily estimated by applying a bootstrapping approach to the CS data, but more advanced approaches have been investigated (Henrys et al., 2015). Alternative options for the prior are possible, and would be worth exploring further to examine sensitivity to the specification of the prior. Where little information is available, an uninformative prior is often used, either uniform, or exponentially declining to capture the parsimony principle that low values of **B** are more likely than high

ones, all else being equal. More usefully, because we iterate over all years independently, we could form the prior distribution at time $t$ from the posterior distribution for the previous year. In practice, we iterate backwards in time, so in fact the posterior at time $t$ becomes the prior for time $t-1$; this is mathematically simple but linguistically confusing. This approach means that information gained in the recent part of the time series is carried over into the earlier part of the time series. Subsequent estimates "borrow strength" from previous ones, in the Bayesian terminology. Currently, we do not use this approach because of the extra computation time this incurs, but methods to speed up this step can be explored.

Observational error can be difficult to estimate objectively and accurately, and often the $\sigma$ terms are poorly known. Even in relative terms, it can be hard to judge the degree of certainty to place in different data sources, where observational error is not readily quantified. In our case, we need to estimate the $\sigma$ terms in the likelihood function (equations 5 - 7) for the AC, EAC and IACS data. Spatial coverage in the data sets is similarly large so there is no clear *a priori* reason to trust one more than the other. However, there are reasons to prioritise the national-scale trends in AC over those from IACS, and to be cautious of the spatial patterns in EAC. AC is a long-established survey with relatively consistent methods, whereas IACS is a recent introduction, and the recording methodology has not been entirely stable over this period (for example, with changes to how much farm woodland is recorded). It also attempts to collect a much higher level of detail (at the individual field scale), and this brings more potential for misclassification to appear as ostensible land-use change. However, with the limited information available, we cannot rule out that this is the more accurate data set, and that EAC and CS underestimate gross change. The accuracy of spatial information in EAC is limited by the way in which the data are collated, using postcodes of the land owner who completes the census return. Where large estates are owned, the correspondence between the centroid of the postcode district and the actual location of the land may not be very close. We therefore ascribe lowest uncertainty to AC, and higher but equal uncertainty to EAC and IACS data. In our Bayesian data assimilation procedure, IACS-based estimates

of **B** are effectively down-weighted when they produce a mismatch with the national-scale AC trends. IACS coverage on forest, urban and other land is not large, and we would not expect accurate detection of changes in these land uses.

A potential problem with the method as we have implemented it is the assumption of independence of errors in the likelihood functions (equations 5 - 7). However, we do not think this is a serious issue here, for the following reasons. Several data sources were used, so different independent estimates of the area of the different land uses are brought in, which mitigates the problem. In all the likelihood functions, $\sigma$ is generally large, making non-independence less of an issue, at least in relative terms. The consequence of assuming non-independence of errors would be to produce unreasonably small uncertainties in the posterior parameters, and this is not the case here.

One of the main problems in land-use studies is that of classification. Depending on definitions used to delimit land-use classes, quite different areas may be calculated for the same nominal classes, and there is a real problem in combining data from different sources in that we may not be comparing like with like. Here, we minimise this problem by using a relatively coarse land-use classification, with only six classes. This would become more problematic if attempting to distinguish more refined classes. The computation time and difficulty increases with the square of the number of land-use classes, so there may be practical limits to the level of detail in the classification used, especially if applying on larger spatial domains.

An attractive feature of the Bayesian data assimilation approach is that additional data sources can be added to the process as they become available, without any major changes to software or step-changes in results. Several other data sources exist in the UK which could be incorporated. These include spatial data on the granting of woodland felling licenses, which would further constrain the likely location of deforestation, and national mapping agency data on urban expansion. As new satellite instruments come on-stream (e.g. from Sentinel and synthetic aperture radar), further remotely-sensed data products will become available

which could be added into the estimation of **A**, **B** and **U**. In this study, we do not attempt to forecast future land-use change, but in principle this is simple with this methodology. If no new data are available, the posterior distribution will widen as future years are iterated over. If scenario data were supplied, such as projected forest planting rates ($G$) or cropland areas required for food security ($A$), these could be used in the estimation of **A**, **B** and **U** in the same way as historical data. The method has applications in providing estimates of historical land use and land-use change input data for modelling work in many domains, including climate modelling (Lawrence et al., 2016), ecosystem and biogeochemical modelling (Ogle et al., 2003; Ostle et al., 2009), species distribution modelling (Dainese et al., 2017; Martin et al., 2013), and socio-economics (Moran et al., 2011; Sharmina et al., 2016).

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
