# Peer review of "Estimation of gross land-use change and its uncertainty using a Bayesian data assimilation approach"

_Biogeosciences, 2017_

## Referee Comment (RC1) · Anonymous Referee #1 · 8 Dec 2017

This is an interesting paper which tackles an important problem, and does so in an interesting way. I have some issues which I think need addressing, and one suggestion.

1. Notation. I do not like the use of upper-case bold letters for vectors, these are usually reserved for matrices, and I think that, in this paper where both vectors and matrices are used this distinction would help with clarity.

2. Notation (ii). Similarly, I do not like the use of upper case U to indicate a scalar value, usually an upper case letter denotes a random variable.

3. I have some concerns about the use of the Countryside Survey data. There is a reference to a bootstrapping procedure, but this is in an inaccessible internal

report which, as far as we can tell, has not been peer-reviewed. What is the bootstrapping seeking to achieve, and how is the stratification which underlies the CS survey accounted for?

4. It is noted later in the paper that 'an increase in one land use has to be balanced by a decrease in one or more other land uses.' Given this I am not convinced by the implicit assumption of independence of errors in the likelihood functions where the densities are simply multiplied together over times and land uses (e.g. Equation 5). I would like to see an explicit defence of this assumption.

5. It would help the author's case if they could use their modelling framework to explore, independently of their data, the scope for variation in $CO_2$ fluxes associated with some fixed net land use change when gross land use changes are varying.

---

## Referee Comment (RC2) · Anonymous Referee #2 · 14 Dec 2017

In "Estimation of land-use change using a Bayesian data assimilation approach" the authors present a Scottish test-case to demonstrate how Bayesian data assimilation could help to more objectively merge different data sources into a single land-use product (with uncertainty intervals on the gross changes). The method is well explained, the test-case is well presented and the discussion nicely balances the strengths and limitations of the method. I have few minor suggestions:

Figure 4 should be improved. In the current figure the CI dominate the signal making it uninformative to show the prior and observations. If that is the message of this figure, then search for a more elegant way to show it (could be a table). The current presentation already uses different ranges of the Y-axis but even then for some rows of subplots the range is not completely used.

[Figure]

Add a chart showing the flow of the method and linking the flow to the different sections on the text. Even the authors did a good job in explaining the flow, it is nice to get a visual overview of the method/manuscript before delving into the details.

I strongly suggest to change the title. The novelty is not in estimating land-use change (actually this study did not estimate land use change at all. It makes use of existing estimates), the novelty is in combining different sources in a reproducible and more objective way. The title should mention the following elements: (1) gross land use changes, (2) combining different data sources into a single product, and (3) uncertainty intervals on the product. If there is some space left you could mention that the approach was Bayesian.

Specific comments: - L 318 replace "sample sample" by "sample" - Fig 2 and 11 change the units of latitude and longitude to degrees, minutes, seconds.

---

## Author Comment (AC1) · 22 Dec 2017

We thank the referee for their thorough reading of the manuscript. We address their points (*shown in italics*) below.

1. Notation - *use of upper-case bold letters for vectors*
The referee makes a valid point, but there is not a perfect solution. The confusion is that some of the vectors we refer to are subsections of a 3D array or matrix (e.g. $\mathbf{U}_{xy}$). We therefore retain the uppercase bold letters for these, and the subscripts define the dimensions referred to. Where the vectors are not part of an array or matrix, we will use lowercase bold as suggested (the vectors $\mathbf{t}$ for time, $\mathbf{g}$ and $\mathbf{l}$ for gains and losses).

[Figure]

There are also some inconsistencies which we will correct: the scalar $A_{gridcell}$ in Eqn 1 should appear as lowercase, so we replace it with $l^2$ where $l$ is the length of the side of the grid cell.

2. Notation - *use of upper case U to indicate a scalar value* rather than *a random variable*.
This convention exists in the stats literature, but we can't try to be consistent with two different conventions. Apart from the points made in (1) above, we think it clearer to use the wider maths convention of uppercase bold denoting matrices (and arrays), with italics denoting the individual elements.

3. Countryside Survey *There is a reference to a bootstrapping procedure, but this is in an inaccessible internal report which, as far as we can tell, has not been peer-reviewed. What is the bootstrapping seeking to achieve, and how is the stratification which underlies the CS survey accounted for?*

Unfortunately, the details of the Countryside Survey data analysis are not available in a peer-reviewed published paper. The 1990 ITE Land Classification (https://catalogue.ceh.ac.uk/documents/235c42f5-6281-40f6-a74c-1b4eb29c78b1) was used to stratify the survey squares, and land-use change was estimated separately for each of the 32 classes. The bootstrapping is attempting to provide confidence intervals on the national-scale estimates of the areas of land-use transition (i.e. the $\mathbf{B}$ matrix). It does this by resampling the data within each class, allowing the within-class variability and classification errors to be propagated. We can add further details to the text, but we are proposing a better method, so CS is not the focus of the paper.

4. *Assumption of independence of errors in the likelihood functions*
In principle, the referee is correct, and the non-independence of the different land uses should be accounted for, rather than summing independent Gaussian terms. However, we do not think this is a serious issue here, for the following reasons.

- Several data sources were used, so different independent estimates of the area of the different land uses are brought in, which mitigates the problem.

- In all the likelihood functions, $\sigma$ is generally large, making non-independence less of an issue, at least in relative terms.

- The consequence of assuming non-independence of errors would be to produce unreasonably small uncertainties in the posterior parameters. We don't see that.

- Unless the referee can see one, there is no obvious way to account for the non-independence mathematically. The Dirichlet distribution has been applied to related problems, where fractions must sum to 1, so the components are intrinsically correlated. However, it is not obvious how this could be applied here, and the method usually fails for numerical computation reasons when dealing with very small numbers. We can add some discussion to this effect, but we don't see an immediate solution.

5. *It would help the author's case if they could use their modelling framework to explore, independently of their data, the scope for variation in CO$_2$ fluxes associated with some fixed net land use change when gross land use changes are varying.*
We did consider this, but the problem seemed to us that if we devised an arbitrary land use change scenario (small fixed net change, larger gross change), the results (the CO$_2$ flux) would be also be arbitrary. A non-arbitrary scenario isn't obvious to us (but open to suggestions). We have submitted a paper elsewhere on the IACS data itself, where we contrast the CO$_2$ fluxes calculated using the detailed gross change versus the CO$_2$ fluxes calculated using only the net change. There is not an analagous comparison here.

---

## Author Comment (AC2) · 22 Dec 2017

We thank the referee for their thorough reading of the manuscript. We address their points (*shown in italics*) below.

*Figure 4 should be improved. In the current figure the CI dominate the signal making it uninformative to show the prior and observations. If that is the message of this figure, then search for a more elegant way to show it (could be a table). The current presentation already uses different ranges of the Y-axis but even then for some rows of subplots the range is not completely used.*
We tried several ways of presenting these data, and we're not sure there is a better

alternative. Firstly, it is helpful to have the figures consistent, and currently Figure 3-6 all have the same form (axes and colour scheme). Each row can have a different y scale, but it becomes messy to re-scale each individual plot. So the scale is set by whatever is largest in a row - either observastions or confidence intervals. The figure does correctly show the relative uncertainties in a form consistent with the other figures, even if some are too small (relatively) to be seen in detail. A table version would be very large and not visually helpful, though this could go in supplementary material.

*Add a chart showing the flow of the method and linking the flow to the different sections on the text.*
We agree this would be a useful addition, and will add this in a revised version.

*I strongly suggest to change the title. The novelty is not in estimating land-use change (actually this study did not estimate land use change at all. It makes use of existing estimates), the novelty is in combining different sources in a reproducible and more objective way. The title should mention the following elements: (1) gross land use changes, (2) combining different data sources into a single product, and (3) uncertainty intervals on the product. If there is some space left you could mention that the approach was Bayesian.*

Our preference would be to leave the title as it is, but we're open to suggestions.

1. We can add the word "gross", but not sure it is really necessary.

2. The term "data assimilation" pretty much captures the idea of *combining different data sources into a single product*.

3. The word "Bayesian" conveys that we are dealing with uncertainty, though perhaps only to the cognisant.

"Estimation of gross land-use change and its uncertainty using a Bayesian data assimilation approach" would be a reasonable compromise.

*Specific comments: - L 318 replace "sample sample" by "sample"*
We will correct this.

*Fig 2 and 11 change the units of latitude and longitude to degrees, minutes, seconds.*
The maps are in British National Grid, so the units are metres east and north of a defined origin. We will clarify this in the caption.

---

## Author Response (AR1)

**Response to Referee Comments**

Peter E. Levy

February 2, 2018

We thank the referee for their thorough reading of the manuscript. We have addressed their points (*shown in italics*) below and revised the manuscript accordingly.

**1 Response to Referee 1**

1. Notation - use of upper-case bold letters for vectors

Although we stated this on line 89, in fact we don't actually refer to any vectors, except where these are subsections of a matrix or 3D array (e.g.  $\mathbf{U}_{xy}$ ). We have removed the word "vectors" from this sentence. We retain the uppercase bold letters for matrices or 3D array, and the subscripts make it clear which dimensions are referred to. There were some inconsistencies which we have corrected: the scalar  $A_{gridcell}$  should appear as lowercase, and could be confused with  $A_{ut}$ , so we replace it with  $l^2$  where l is the length of the side of the grid cell. Also, the array w should appear as uppercase, which we have corrected.

2. Notation - use of upper case U to indicate a scalar value rather than a random variable.

This convention exists in the stats literature, but we can't be consistent with two different conventions. We think it clearer to use the wider maths convention of uppercase bold denoting matrices (and arrays), with italics denoting the individual elements.

3. Countryside Survey There is a reference to a bootstrapping procedure, but this is in an inaccessible internal report which, as far as we can tell, has not been peer-reviewed. What is the bootstrapping seeking to achieve, and how is the stratification which underlies the CS survey accounted for?

Unfortunately, the details of the Countryside Survey data analysis are not available in a peer-reviewed published paper. The 1990 ITE Land Classification (https://catalogue.ceh.ac.uk/documents/235c42f5-6281-40f6-a74c-1b4eb29c78b1) was used to stratify the survey squares, and land-use change was estimated separately for each of the 32 classes. The bootstrapping is attempting to provide confidence intervals on the national-scale estimates of the areas of land-use transition (i.e. the **B** matrix). It does this by resampling the data within each class, allowing the within-class variability and classification errors to be propagated. We have added further details to the text, but we are proposing a better method, so CS is not the focus of the paper.

**4. Assumption of independence of errors in the likelihood functions**

In principle, the referee is correct, and the non-independence of the different land uses should be accounted for, rather than summing independent Gaussian terms. However, we do not think this is a serious issue here, for the following reasons, which we have added to the text in the Discussion.

- Several data sources were used, so different independent estimates of the area of the different land uses are brought in, which mitigates the problem.
- In all the likelihood functions,  $\sigma$  is generally large, making non-independence less of an issue, at least in relative terms.
- The consequence of assuming non-independence of errors would be to produce unreasonably small uncertainties in the posterior parameters. We don't see that.
- Unless the referee can see one, there is no obvious way to account for the non-independence mathematically. The Dirichlet distribution has been applied to related problems, where fractions must sum to 1, so the components are intrinsically correlated. However, it is not obvious how this could be applied here, and the method usually fails for numerical computation reasons when dealing with very small numbers. We can add some discussion to this effect, but we don't see an immediate solution.

5. It would help the authors case if they could use their modelling framework to explore, independently of their data, the scope for variation in  $CO_2$  fluxes associated with some fixed net land use change when gross land use changes are varying.

We did consider this, but the problem seemed to us that if we devised an arbitrary land use change scenario (small fixed net change, larger gross change), the results (the  $CO_2$  flux) would be also be arbitrary. A non-arbitrary scenario isn't obvious to us (and none was suggested in the open discussion process). We have submitted a paper elsewhere on the IACS data itself, where we contrast the  $CO_2$  fluxes calculated using the detailed gross change versus the  $CO_2$  fluxes calculated using the net change. There is not an analogous comparison here.

**2 Response to Referee 2**

Figure 4 should be improved. In the current figure the CI dominate the signal making it uninformative to show the prior and observations. If that is the message of this figure, then search for a more elegant way to show it (could be a table). The current presentation already uses different ranges of the Y-axis but even then for some rows of subplots the range is not completely used.

We tried several ways of presenting these data, and we're not sure there is a better alternative. Firstly, it is helpful to have the figures consistent, and currently Figure 3-6 all have the same form (axes and colour scheme). Each row can have a different y scale, but it becomes messy to re-scale each individual plot. So the scale is set by whatever is largest in a row - either observations or confidence intervals. The figure does correctly show the relative uncertainties in a form consistent with the other figures, even if some are too small (relatively) to be seen in detail. A table version would be very large and not visually helpful.

Add a chart showing the flow of the method and linking the flow to the different sections on the text.

We have added this as a new Figure 2 in the revised version.

I strongly suggest to change the title. The novelty is not in estimating landuse change (actually this study did not estimate land use change at all. It makes use of existing estimates), the novelty is in combining different sources in a reproducible and more objective way. The title should mention the following elements: (1) gross land use changes, (2) combining different data sources into a single product, and (3) uncertainty intervals on the product. If there is some space left you could mention that the approach was Bayesian.

We contend that the paper is about estimating land-use change, but may be this is just semantics.

- 1. We add the word "gross", to make this aspect explicit.
- 2. The term "data assimilation" pretty much captures the idea of *combining* different data sources into a single product.
- 3. The word "Bayesian" conveys that we are dealing with uncertainty, though perhaps only to the cognisant.

so "Estimation of gross land-use change and its uncertainty using a Bayesian data assimilation approach" seems a reasonable compromise.

Specific comments: - L 318 replace sample sample by sample We have corrected this.

Fig 2 and 11 change the units of latitude and longitude to degrees, minutes, seconds.

The maps are in British National Grid, so the units are metres east and north of a defined origin. We have clarified this in both captions.

Additionally, we have now used the correct Copernicus Citation Style Language file, and reference citations and bibliography have changed accordingly, but should all now be correct.

**Estimation of gross land-use change and its uncertainty using a Bayesian data assimilation approach**

4 Peter Levy, Marcel van Oijen, Gwen Buys, and Sam Tomlinson

2018-02-02

**6 Abstract**

5

We present a method for estimating land-use change using a Bayesian data assimilation 7 approach. The approach provides a general framework for combining multiple disparate data 8 sources with a simple model. This allows us to constrain estimates of gross land-use change 9 with reliable national-scale census data, whilst retaining the detailed information available 10 from several other sources. Eight different data sources, with three different data structures, 11 were combined in our posterior estimate of land-use and land-use change, and other data 12 sources could easily be added in future. The tendency for observations to underestimate 13 gross land-use change is accounted for by allowing for a skewed distribution in the likelihood 14 function. The data structure produced has high temporal and spatial resolution, and is 15 appropriate for dynamic process-based modelling. Uncertainty is propagated appropriately 16 into the output, so we have a full posterior distribution of output and parameters. The 17 data are available in the widely used netCDF file format from http://eidc.ceh.ac.uk/ (doi 18 pending). 19

20

**21 Introduction**

Human-induced land-use change has a substantial impact on biodiversity and both biogeo-22 chemical and hydrological cycles (Post & Kwon, 2000; Gitz & Gitz and Ciais, 2003; Levy et 23 al. et al., 2004; Newbold et al. et al., 2015; Piano et al. 
[revised manuscript text omitted]
  | $\mathbf{U},\mathbf{B},rac{\mathbf{w}}{\mathbf{W}}$               | 2004-2015                 |
|              | and Control System         |                                                                    |                           |
| NFEW         | FC National Forest Estate  | $\mathbf{U},\mathbf{B},rac{\mathbf{w}}{\mathbf{W}}$               | 1969-2014                 |
|              | and Woodlands              |                                                                    |                           |
| FC           | FC new planting            | $\mathbf{G}_{	ext{forest}}$                                        | 1969-2016                 |
| LCM          | CEH Land Cover Map         | $\mathbf{A}_{\mathrm{urban}},\mathbf{U},\overline{\boldsymbol{w}}$ | 1990, 2000, 2007, 2015    |
| ALCM         | Agricultural Land Capabil- | $w_{\overline{\mathbf{W}}}$                                        | NA                        |
|              | ity Map                    |                                                                    |                           |

Table 1: Data sources assimilated in the estimation of land-use change in Scotland.

**147 Data assimilation**

148 Our data assimilation method is represented graphically in 2 and proceeded as follows.

• From repeat ground-based surveys, the CEH Countryside Survey (CS, Norton *et al.*et

Figure 2: Schematic diagram showing information flow in the data assimilation procedure. Data sources are listed in Table 1. The prior estimate of the transition matrix  $\mathbf{B}$  at each time point is provided by the CEH Countryside Survey (CS). Observations of the area (A) occupied by each land use type u, the gross gains and losses (G and L), and spatially-explicit estimate of land use  $(\mathbf{U}^{obs})$  are combined in a Bayesian calibration via the likelihood functions (equations 5 - 7) to produce updated, posterior estimates of the transition matrix  $\mathbf{B}^{\text{post}}$ . We then use spatial and probabilistic information on the location of land-use change  $(\mathbf{W})$  to simulate posterior realisations of land use and land-use change  $(\mathbf{U}^{\text{post}})$ .

al., 2012; Wood *et al.*, 2017) provides direct observations of **B** for approximately 150 150 1-km2 survey squares in Scotland. Whilst the coverage is not large compared to 151 the total area of Scotland, the sample squares were chosen on a stratified design, and 152 the observations are valuable in having consistent recording methods over a long time 153 period. The method for scaling these survey squares to national scale is described 154 in (Milne & and Brown, 1997). Surveys were carried out in 1978, 1984, 1990, 2000, 155 and 2007, and we interpolated linearly between survey years to produce an annual 156 time series. We used the estimates derived in this way as our prior distribution of **B**. 157 Each year, the mean of the prior distribution was taken to be the value of **B** from 158 CS. The standard deviation  $\sigma$  of the prior distribution was estimated by applying 159 a bootstrapping approach from an earlier bootstrapping approach applied to the CS 160 data (Scott, 2008), in an attempt to provide confidence intervals on the national-scale 161 estimates of the areas of land-use transition (i.e. the **B** matrix). 162

• National Agricultural Census (AC) data provide annual records of the total area in 163 the main agricultural land uses (Scottish Government, 2017). The Agricultural Census 164 is conducted in June each year by the government agriculture department. Farmers 165 declare the agricultural activity on their land in the form of ca. 150 items of data via a 166 postal questionnaire. The results are collated at national scale. These are a long-running 167 data set with near-complete coverage of agricultural land, relatively consistent over 168 time, and are reported as national statistics and to the FAO. Hence it is desirable for 169 our estimates of land-use change to be consistent with these data as far as possible. We 170 therefore use these data as observations of  $A_{ut}$  in the Bayesian framework, and predict 171  $\Delta A_{ut}$  from  $\mathbf{B}_t$  according to equation 4. The likelihood of the net change observed by 172 Agricultural Census ( $\Delta A_{ut}^{obs}$ ) arising from normal distributions with means determined 173 by equation 4 and the parameter matrix  $\mathbf{B}$  is 174

$$\mathcal{L}_{\text{net}} = \prod_{\substack{u=1\\t=1}}^{n_u} \frac{1}{\sigma_{ut}^{\text{obs}} \sqrt{2\pi}} \exp(-(\Delta A_{ut}^{\text{obs}} - \Delta A_{ut}^{\text{pred}})^2 / 2\sigma_{ut}^{\text{obs}^2})$$
(5)

where  $\Delta A_{ut}^{\text{pred}}$  is the prediction from equation 4 for the change in land use u at time t, and 175  $\sigma_{ut}^{obs}$  is the observational error in the Agricultural Census. So, we now have (i) a simple model 176 which predicts net land-use change in terms of a parameter matrix; (ii) prior estimates of 177 these parameters for each year from the Countryside Survey; and (iii) a function (equation 5) 178 for the likelihood of the observations of net change given the model parameters. Combining 179 these in Bayes Theorem, we can estimate the posterior distribution of the parameters, the 180 transition matrix **B**. However before describing this, we can extend this simplest likelihood 181 function by adding further sources of observational data. 182

• The EDINA Agricultural Census (EAC) data (http://agcensus.edina.ac.uk/) provide 183 additional information on land-use change, as they attempt to produce a spatially explicit 184 version of the national-scale Agricultural Census data. Farm-level data is aggregated 185 to 2-km grid cells, and data are available (or can be inferred) annually. While not 186 containing explicit information on the actual land-use transitions, the resolution of the 187 data is high enough that the net changes recorded each year in each 2-km cell may 188 approximate the gross changes. In other words, because the data records the annual 189 increases and decreases in land use across the grid of 2-km cells, the national totals of 190 these increases and decreases gives an estimate of the gross change, the row and column 191 sums of the transition matrix  $\mathbf{B}$ , as well as the net change. When calculating the 192 likelihood in our Bayesian framework, we can thus use the more informative observations 193 of gross gains and losses (G and L) rather than just the observations of net change 194  $(\Delta \mathbf{A})$  from the national Agricultural Census. However, we know that the observations 195 will tend to underestimate the gross change, because of the nature of the data reporting 196 process: any counter-balancing gross change within the 2-km square is not included. To 197 account for this, we can use a skewed normal distribution to represent this, such that 198

predictions which overestimate the observations are more likely than underestimates. 199 A skewed normal distribution of this form (Azzalini, 2017) gives the likelihood of the 200 gross changes observed as: 201

$$\mathcal{L}_{\text{gross}} = \prod_{\substack{u=1\\t=1}}^{n_u} \frac{2}{\sigma_{L_{ut}^{\text{obs}}}} \phi\left(\frac{L_{ut}^{\text{obs}} - L_{ut}^{\text{pred}}}{\sigma_{L_{ut}^{\text{obs}}}}\right) \Phi\left(\alpha\left(\frac{L_{ut}^{\text{obs}} - L_{ut}^{\text{pred}}}{\sigma_{L_{ut}^{\text{obs}}}}\right)\right) \\ \times \frac{2}{\sigma_{G_{ut}^{\text{obs}}}} \phi\left(\frac{G_{ut}^{\text{obs}} - G_{ut}^{\text{pred}}}{\sigma_{G_{ut}^{\text{obs}}}}\right) \Phi\left(\alpha\left(\frac{G_{ut}^{\text{obs}} - G_{ut}^{\text{pred}}}{\sigma_{G_{ut}^{\text{obs}}}}\right)\right)$$
(6)

where  $\phi$  is the standard normal probability density function,  $\Phi$  is the corresponding cumulative 202 density function, and  $\alpha$  is the skew parameter. Positive  $\alpha$  produces a positive skew (when 203  $\alpha = 0$  we have the standard normal distribution). The parameter  $\alpha$  can itself be estimated 204 as part of the data assimilation procedure. 205

- Several data sources provide observations of U for one or more land uses at a restricted 206 set of time points. We combine these into a single array  $\mathbf{U}^{\text{obs}}$  as follows. 207
- For an initial estimate of U, we use the Corine data sets for 1990, 2000, 2007, and 208 2012 (European Environment Agency, 2016). For each grid cell, change between 209 these years was assumed to occur at a random time within the interval, so that at 210 national scale we effectively interpolate linearly. This produces U with complete 211 UK coverage at annual resolution over the period 1990 to 2012. 212
- We overlay this with IACS data over the period 2004 to 2015 (Tomlinson *et* 213 al. et al., 2017). The Integrated Administration and Control System (IACS) is 214 a European-wide spatially explicit dataset at the field level that serves as a 215 register of agricultural subsidy claims under the EU Common Agricultural Policy. 216 IACS records field-level land use (crop type, grassland age, forest coverage), field 217 geometry and its association to a farm holding. This has large, but not complete 218 spatial coverage (65 % of the Scottish land area), and the Corine data are retained 219

where IACS data are missing. Where there are conflicts with Corine, IACS data are given precedence because they are direct ground-based records.

222 – We then add forestry data from the GB Forestry Commission (FC) National 223 Forest Estate and Woodlands (https://www.forestry.gov.uk/datadownload), which 224 records the location and planting date of forestry. Again, this only has limited 225 coverage, as it only covers forest land, but is given precedence in the case of conflict 226 with the Corine/IACS data. We iterate over each time step to calculate  $\mathbf{B}_t^{\text{obs}}$  with 227 equation 3.  $\mathbf{B}_t^{\text{obs}}$  thus contains an observed estimate of the transition matrix for 228 each year, from the combination of Corine, IACS and FC data.

We can therefore add an additional term to the likelihood function which incorporates the comparison of the observations  $\mathbf{B}^{\text{obs}}$  with the values in the current parameter set  $\mathbf{B}^{\text{pred}}$ .

$$\mathcal{L}_{\mathbf{B}} = \prod_{\substack{i=1\\j=1\\j=1\\t=1}}^{n_u} \frac{1}{\sigma_{\beta_{ijt}^{\text{obs}}} \sqrt{2\pi}} \exp(-(\beta_{ijt}^{\text{obs}} - \beta_{ijt}^{\text{pred}})^2 / 2\sigma_{\beta_{ijt}^{\text{obs}}}^2)$$
(7)

• To establish the posterior distribution, we use the Markov Chain Monte Carlo (MCMC) 232 approach with the "DEz" algorithm implemented in the R package BayesianTools 233 (Hartig *et al.*, 2017). For each interval in the 46 year time series, an MCMC 234 simulation was run, using the prior  $\mathbf{B}_t$  matrix from Countryside Survey, the observations 235 of  $\Delta \mathbf{A}_t$ ,  $\mathbf{L}_t$ ,  $\mathbf{G}_t$  for that year, and the observed  $\mathbf{B}_t$  matrix from Corine-IACS\_NFEW. In 236 practice, it is more convenient to use log-likelihoods, and our overall likelihood was the 237 summation of  $\log(\mathcal{L}_{net})$ ,  $\log(\mathcal{L}_{gross})$  and  $\log(\mathcal{L}_{B})$ . Nine chains were used, with 100,000 238 interations in each. To establish the initial **B** parameter values for one of the chains, a 239 least-squares fit with the  $\Delta \mathbf{A}$  was used. Other chains were over-dispersed by adding 240 random variation to this best-fit parameter set. 241

• Having established the posterior distribution of  $\mathbf{B}$ , we use spatial and probabilistic

information on the location of land-use change to simulate posterior realisations of 243  $\mathbf{U}^{\text{post}}$ . Starting with our best estimate of the near-present state of land use,  $\mathbf{U}_{t=2015}^{\text{obs}}$ , 244 we work backwards in time. At each time step, we know the number of grid cells which 245 need to change from land use i to land use j from the posterior matrix  $\mathbf{B}_t$ . For each i 246 to j transition, we perform a weighted sampling operation to select this number of cells 247 from those where  $U_{xyt} = i$ . In choosing which cells to assign to j, we use the available 248 data to calculate the probabilities which weight the sampling. Recall that  $\mathbf{U}^{\text{obs}}$  is given 249 by the amalgamation of Corine, IACS and NFEW data. In the simplest case, the 250 probabilities are determined only by this: all cells where  $U_{xyt}^{obs} = i$  and  $U_{xy,t-1}^{obs} = j$  have 251 equally high probability of being selected in the sample, and all cells where  $U_{xyt}^{obs} = i$ 252 and  $U_{xy,t-1}^{obs} \neq j$  have equally low (but non-zero) probability of being selected in the 253 sample. This requires only a few simple rules to construct the probability weightings, 254 w, for sampling cells for conversion from *i* to *j*: 255

if
$$U_{xy,t}^{\text{obs}} \neq i$$
 then  $\underline{w}W_{xy} \leftarrow 0$  else  $\underline{w}W_{xy} \leftarrow 1$
 $\land$  if  $U_{xy,t-1}^{\text{obs}} = j$  then  $\underline{w}W_{xy} \leftarrow 1$  else  $\underline{w}W_{xy} \leftarrow p_m$

where  $p_m$  is the probability of cells being misclassified in  $\mathbf{U}^{\text{obs}}$ , which we estimate to be 256 0.05. Sampling is done without replacement, so that a grid cell can only be selected 257 once per year. To illustrate with an exa